# Endocytosis at the *Drosophila* blood–brain barrier as a function for sleep

**Gregory Artiushin[1], Shirley L Zhang[2], Hervé Tricoire[3], Amita Sehgal[1,2]\***

[1]Neuroscience Graduate Group, Perelman School of Medicine, University of Pennsylvania, Philadelphia, United States; [2]Howard Hughes Medical Institute, Perelman School of Medicine, University of Pennsylvania, Philadelphia, United States; [3]Laboratory of Degenerative Processes, Stress and Aging, UMR8251, Université Paris Diderot, Paris, France

**Abstract** Glia are important modulators of neural activity, yet few studies link glia to sleep regulation. We find that blocking activity of the endocytosis protein, dynamin, in adult *Drosophila* glia increases sleep and enhances sleep need, manifest as resistance to sleep deprivation. Surface glia comprising the fly equivalent of the blood-brain barrier (BBB) mediate the effect of dynamin on sleep. Blocking dynamin in the surface glia causes ultrastructural changes, albeit without compromising the integrity of the barrier. Supporting a role for endocytic trafficking in sleep, a screen of Rab GTPases identifies sleep-modulating effects of the recycling endosome Rab11 in surface glia. We also find that endocytosis is increased in BBB glia during sleep and reflects sleep need. We propose that endocytic trafficking through the BBB represents a function of sleep.
DOI: https://doi.org/10.7554/eLife.43326.001

## Introduction

Sleep is a conserved behavioral state of fundamental significance. Nevertheless, the nature of its relevance to brain function is still a matter of active research and debate (*Krueger et al., 2016*). Most studies of sleep function focus on consequences of sleep loss for performance, in particular cognitive ability, or for overall physiology. Little is known about the impact of sleep on basic cell physiology or the extent to which cellular perturbations affect sleep. Amid the existing hypotheses for the purpose of sleep, multiple avenues exist for glial contribution to both regulation and function (*Frank, 2018*), which have been minimally explored.

Glial function has been linked to sleep in some contexts (*Bjorness et al., 2016*; *Chen et al., 2015*; *Halassa et al., 2009*; *Farca Luna et al., 2017*; *Seugnet et al., 2011*). Sleep is well-associated with improved learning and memory, and appears to support synaptic remodeling (*Krueger et al., 2016*; *Tononi and Cirelli, 2014*), which may be accomplished in part by astrocyte and microglial activation (*Bellesi et al., 2017*). Clearance of brain interstitial fluid has been shown to require the glymphatic system that involves astrocytes and whose function is enhanced during sleep (*Xie et al., 2013*). Glial signaling may also contribute to sleep need (*Bjorness et al., 2016*; *Halassa et al., 2009*), potentially as a consequence of astrocytic sensing of metabolic or energetic conditions (*Clasadonte et al., 2017*).

In *Drosophila* (*Seugnet et al., 2011*) and mice (*Halassa et al., 2009*), astrocytes reportedly influence sleep amount or quality following sleep deprivation. Disruption of vesicular trafficking in mouse astrocytes by dnSNARE expression prevented homeostatic rebound sleep following deprivation, suggesting that glia release sleep-promoting factors (*Halassa et al., 2009*). Likewise, *Notch* signaling acts in fly astrocytes to modulate sleep and learning in response to sleep loss (*Seugnet et al., 2011*). Endocytic/exocytic trafficking is also important in fly glia for the regulation of circadian behavior (*Ng and Jackson, 2015*; *Ng et al., 2011*), but effects on sleep have not been explored.

**\*For correspondence:**
amita@pennmedicine.upenn.edu

**Competing interests:** The authors declare that no competing interests exist.

Given the genetic tools and accessibility of defined glial populations in *Drosophila* (*Edwards and Meinertzhagen, 2010*), the model is well positioned to discover new aspects of glial influence on sleep.

## Results

### Blocking vesicular trafficking in glia increases sleep

To address the role of glia in homeostatic sleep in *Drosophila*, specifically rebound sleep following deprivation, we blocked vesicular trafficking during sleep deprivation using *Shibire*[ts1] (or *Shi*[1]), a temperature-sensitive dominant negative allele of *dynamin*, a GTPase involved in membrane scission (*van der Bliek and Meyerowitz, 1991*). Flies expressing Shibire[ts1] (UAS-*Shi*[1]) by a pan-glial driver (*Repo*-GAL4) were raised at permissive temperature (18°C) until adulthood and subjected to mechanical sleep deprivation while being monitored for sleep/activity levels. Concurrently with the start of mechanical deprivation at lights off (Zeitgeber Time (ZT) 12), *Shi*[1] was induced by raising the temperature to a restrictive 30°C. Mechanical deprivation was applied for 12 hr, concluding at the beginning of the following day (ZT 0), when the temperature was returned to permissive 18°C.

While mechanical sleep deprivation is not always total, and acclimation occurs over longer timespans, Repo-GAL4 > UAS-Shi[1] flies exhibited a surprisingly large amount of sleep above controls throughout the deprivation period, precluding any analysis of subsequent rebound. Experimental flies slept over 300 min in the course of 12 hr of mechanical deprivation, significantly greater than both parental controls (*Figure 1A*). This effect was present in both males and females and was confirmed with an additional line containing an independent single *Shi*[1] insertion (UAS-*20xShi*[1]) (*Figure 1A*).

In principle, *Shi*[1] is an allele for a temperature-sensitive dynamin whose vesicle-forming membrane scission function declines with heightened temperature to act as a dominant negative. However, quite surprisingly, we found that increased sleep evident during mechanical deprivation was not dependent on a shift from permissive to restrictive temperature, as *Repo* > *Shi*[1] flies deprived at 18°C exhibited greater sleep as compared to controls, commensurate to experimental flies at 30°C (*Figure 1B*). As discussed later, the temperature benchmarks established for dynamin inactivation may not be appropriate for all cell populations or phenotypes (*Kilman et al., 2009*). Importantly, the locomotor activity during wake (activity index) of *Repo* > *Shi*[1] flies at permissive temperature does not significantly differ from that of controls (*Figure 2*), suggesting that the effect is particular to sleep and not simply locomotor impairment which may otherwise be observed with glial manipulation (*Zhang et al., 2017*). Additionally, to rule out that *Repo* > *Shi*[1] flies may be less sensitive to the mechanical deprivation stimulus, we stimulated flies between ZT11 and ZT12, a time when most flies are awake. We found that stimulation of awake flies increased beam crossing, and that the average locomotor activity for experimental flies during this time did not differ significantly from that of controls (*Figure 1—figure supplement 1*). Therefore, it is unlikely that the increased sleep of *Repo* > *Shi*[1] flies during mechanical deprivation is a result of diminished sensitivity to the stimulus.

Based on the resistance to sleep deprivation displayed by *Repo* > *Shi*[1] flies, which may represent greater sleep pressure, we closely examined daily, unperturbed sleep. Flies expressing *Shi*[1] in all glia had significantly greater total sleep than controls (*Figure 2A*) at 18°C, which was consistent across independent *Shi*[1] lines (*Figure 2B*) in females as well as males (*Figure 2—figure supplement 1A–B*). Expression of *Shi*[1] with *Repo*-GAL4 at permissive temperature resulted in an equivalent number of sleep bouts across the day, but significantly longer average bout lengths in almost all lines and sexes (see exception in *Figure 2—figure supplement 1A*), suggesting that sleep was also better consolidated (*Figure 2* and *Figure 2—figure supplement 1A–B*).

While we did not observe an appreciable difference in resistance to sleep deprivation between 18° C and 30°C, adult *Repo->Shi*[1] flies raised at 18°C were also shifted to 30°C for 2 days to examine if the baseline sleep phenotype could be exaggerated. The pattern of *Drosophila* sleep across the day is known to alter at temperatures exceeding those preferred by flies (~25°C)– with greater sleep occurring during the daytime at the expense of nighttime sleep (*Parisky et al., 2016*). Appropriately, when kept at 30°C, sleep loss occurred during the nighttime in all flies, but an appreciably greater amount of nighttime sleep was seen in *Repo* > *Shi*[1] flies (*Figure 2—figure supplement 2A*). Total sleep at 30°C in experimental female flies was equivalent to that at 18 degrees (*Figure 2—*

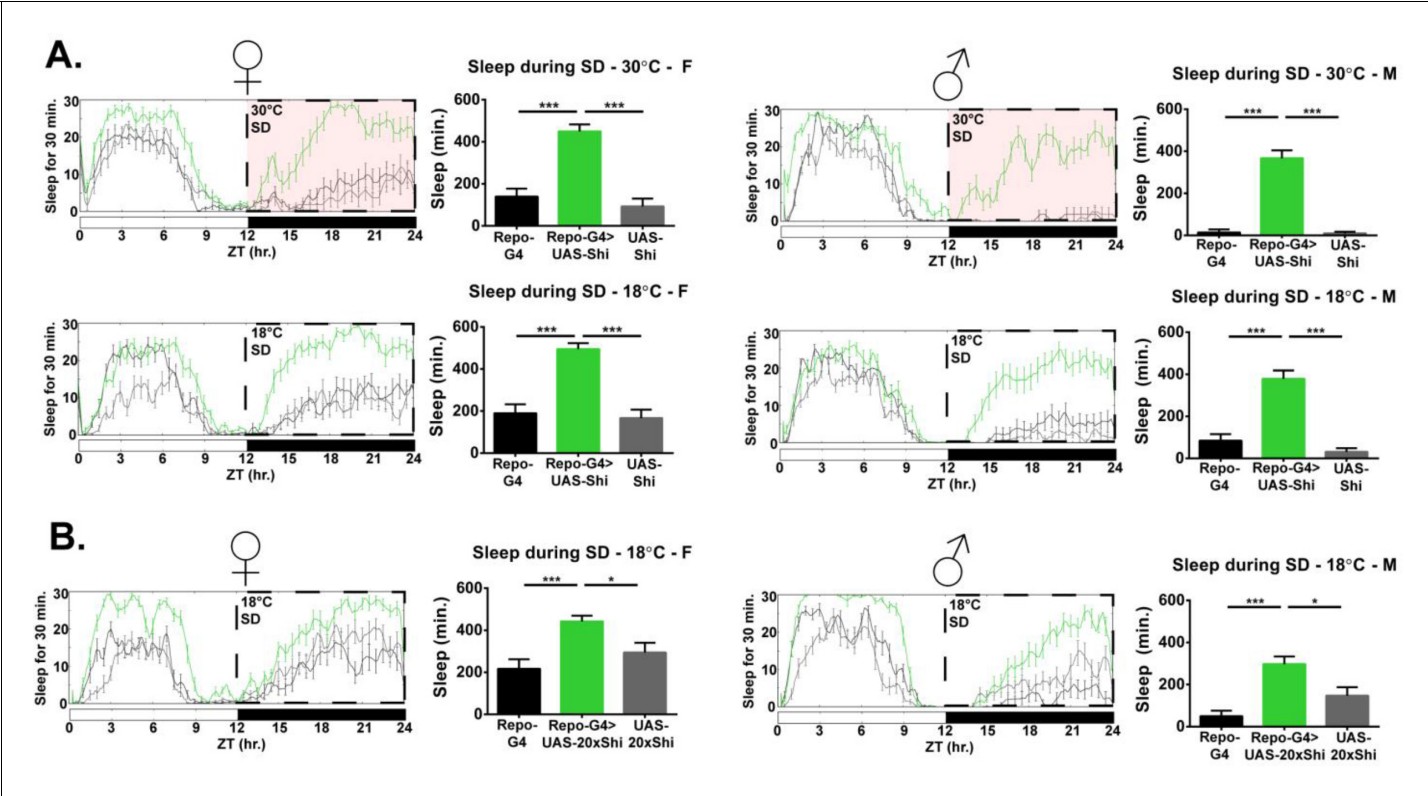

**Figure 1.** Inhibition of endocytosis in glia produces resistance to mechanical sleep deprivation. (**A**) Resistance to mechanical sleep deprivation, manifest as elevated sleep during stimulation, for *Repo*-GAL4 > UAS *Shi.ts1*;UAS-*Shi.ts1* female flies (n = 14–16, per genotype) and male (n = 16, per genotype) at both 30°C and 18°C. Dashed box indicates the period of mechanical deprivation, ZT12-24. Red shading indicates 30°C, otherwise the temperature is 18°C. (**B**) Sleep during deprivation at permissive temperature (18°C) was also present using an additional *Shi[1]* line, in *Repo*-GAL4 > UAS-*20xShi.ts1* females (n = 16, per genotype) and males (n = 16, per genotype). One-way ANOVA with Holm-Sidak post-hoc test, *p<0.05, **p<0.01, ***p<0.001. Error bars represent standard error of the mean (SEM).

DOI: https://doi.org/10.7554/eLife.43326.002

The following figure supplement is available for figure 1:

**Figure supplement 1.** Glial *Shi[1]* expression does not affect responsiveness to mechanical stimulus during wake.

DOI: https://doi.org/10.7554/eLife.43326.003

*figure supplement 2B*, and even diminished in males *Figure 2—figure supplement 2C*), suggesting that the effect of glial *Shi[1]* on daily sleep is substantial even at allegedly permissive temperature. A nighttime phenotype is not seen at 18°C most likely due to ceiling amounts of sleep in controls but is clearly present at 30°C, where the temperature-dependent change in sleep pattern shifts control sleep to daytime hours. This suggests that *Shi[1]* expression can increase sleep throughout the day.

To determine whether increased sleep is the result of a partially dominant-negative *Shi[1]* at lower temperature or the overexpression of *Shi[1]*, we compared the effects of WT dynamin (*Shi[WT]*) with *Shi[K44A]*, a constitutive dominant negative allele of *dynamin*. While *Shi[K44A]* is also deficient for GTP hydrolysis, it is a distinct mutation from *Shi[1]*(G273D) (*van der Bliek and Meyerowitz, 1991*). Expression of *Shi[K44A]* in all glia resulted in significantly elevated sleep, in particular during the day, while expression of *Shi[WT]* had no effect (*Figure 2C*, *Figure 2—figure supplement 3* for total sleep). Given that a constitutively dominant negative dynamin phenocopies the effect of *Shi[1]* expression at 18°C, the sleep phenotype likely arises from inhibition of dynamin by *Shi[1]* even at the reportedly permissive temperature.

## Glia of the blood brain barrier link sleep to vesicular trafficking

As noted above, astrocytes have previously been implicated in sleep (*Bjorness et al., 2016*; *Halassa et al., 2009*; *Seugnet et al., 2011*). In the fly, various glial classes have been described with

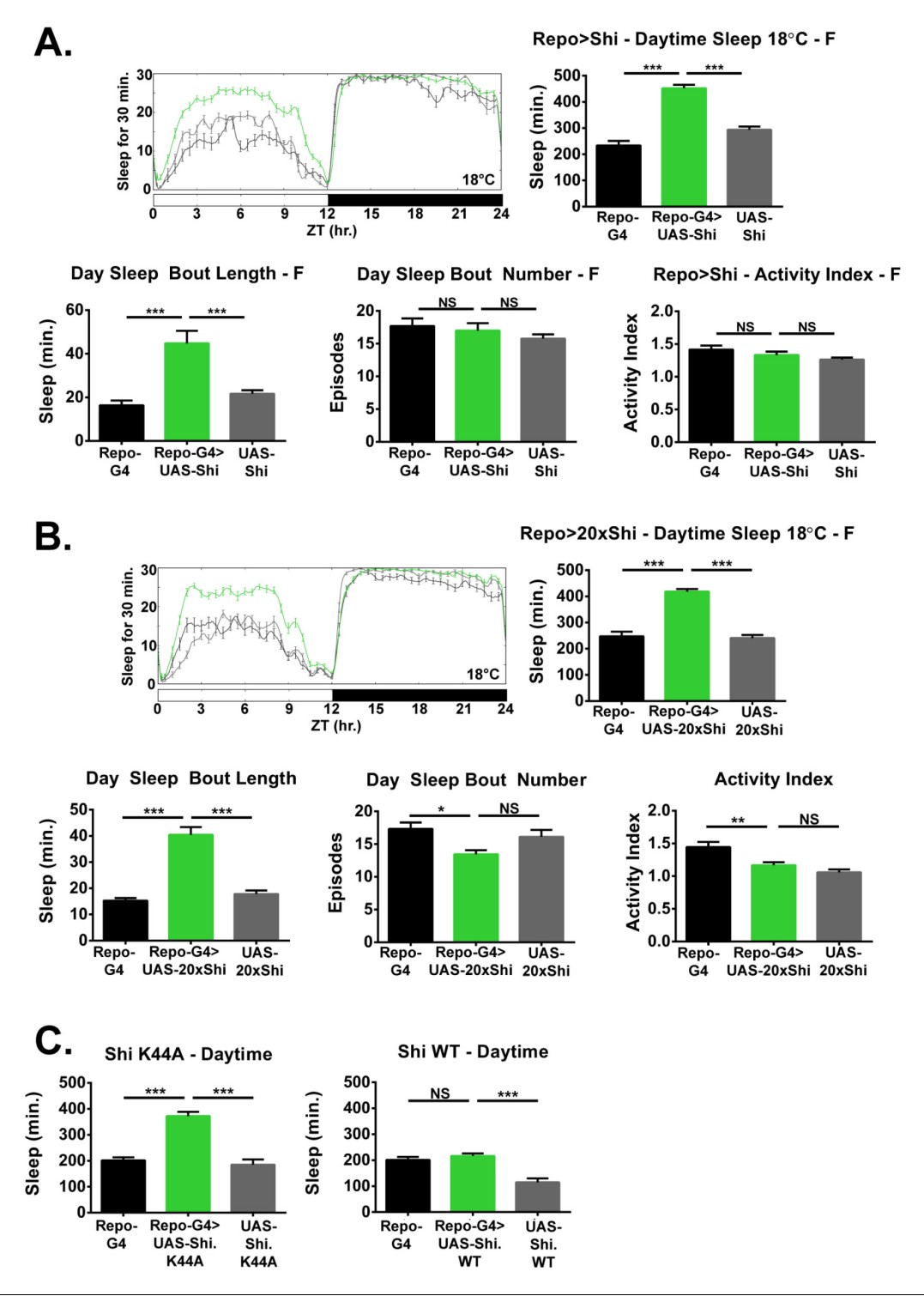

**Figure 2.** Inhibition of endocytosis in glia increases baseline sleep. (A) Daily baseline sleep amount (3 day mean) and sleep/activity characteristics including mean daytime sleep bout number, bout length, and activity index for *Repo*-GAL4 > UAS *Shi.ts1*;UAS-*Shi.ts1* female flies at 18°C (n = 19–26), and (B) *Repo*-GAL4 > UAS-*20xShi.ts1* females (n = 23–31). (C) Daytime sleep for female flies at 18°C expressing either a constitutively dominant negative Shi (UAS-*Shi.K44A*) (n = 15–16) or a wild-type Shi (UAS-*Shi.WT*) (n = 15–32) by *Repo*-GAL4. One-way ANOVA with Holm-Sidak post-hoc test, *p<0.05, **p<0.01, ***p<0.001. Error bars represent standard error of the mean (SEM).

DOI: https://doi.org/10.7554/eLife.43326.004

The following figure supplements are available for figure 2:

*Figure 2 continued on next page*

*Figure 2 continued*

**Figure supplement 1.** Inhibition of endocytosis in glia increases baseline sleep in males.
DOI: https://doi.org/10.7554/eLife.43326.005
**Figure supplement 2.** *Shi[1]* expression at 30°C in glia increases baseline sleep similar to expression at 18°C.
DOI: https://doi.org/10.7554/eLife.43326.006
**Figure supplement 3.** Inhibition of endocytosis in glia increases total baseline sleep.
DOI: https://doi.org/10.7554/eLife.43326.007

corresponding GAL4 drivers generated for these classes (*Stork et al., 2012*). Apart from astrocytes (*Doherty et al., 2009*), *Drosophila* glia include the ensheathing/wrapping glia (*Doherty et al., 2009*; *Ito et al., 1995*), cortex glia (*Awasaki et al., 2008*), and the two layers of the surface or blood-brain barrier glia (*Awasaki et al., 2008*; *Schwabe et al., 2005*). To determine whether any specific glial class is sufficient for the *Shi[1]* phenotype, we expressed *Shi[1]* at the permissive temperature using a panel of previously published GAL4 drivers targeting different glial populations (*Figure 3A*), with preference given to drivers which would isolate individual glial classes. Surprisingly, we did not observe a significant effect on sleep as compared to controls when using drivers expressed in astrocyte-like glia. Instead, expression of *Shi[1]* in the subperineurial (SPG) glia of the blood brain barrier, by *moody*-GAL4, produced a significant increase in total (*Figure 3A*) and daytime sleep *Figure 3—figure supplement 1*. The same was true, although to a lesser extent, with the NP6293-GAL4 driver, which expresses in the perineurial glia (PG), the second layer of the fly BBB (example expression *Figure 3—figure supplement 2*. The cumulative sleep increase produced by the two drivers is somewhat less than with Repo-G4, which likely reflects discrepancy in driver strength, but could also suggest a role for other populations – nevertheless only the surface glial drivers were sufficient in isolation. Surface glia were also sufficient for the resistance to sleep deprivation phenotype, although only through the PG population (*Figure 3—figure supplement 3*Fig., S3.3). Together these data indicate an involvement of BBB glia in *Shi[1]*-mediated effects on fly sleep.

## Inducing block of vesicular trafficking in adult blood brain barrier glia increases sleep

As an additional glial sub-type GAL4 driver, we observed that *Rab9*-GAL4 exhibits prominent expression in the BBB glia. To identify the BBB glial population labeled by *Rab9*-G4 we drove expression of a nuclear-localized GFP, which can differentiate the surface glial populations based on nuclear morphology and abundance (*DeSalvo et al., 2014*). The nuclei of perineurial glial are numerous and relatively small, while those of subperineurial glia are distinctively large and limited in number, consistent with the size and distribution of this glial class. The pattern of expression of nuclear GFP under the control of *Rab9*-G4 was typical of the subperineurial glia (*Figure 3—figure supplement 2A*), marking Rab9-G4 as an SPG driver. In addition, *Rab9*-G4 also shows some, albeit sparse, neuronal expression in the brain (*Figure 3—figure supplement 2B* for *Repo*-Gal80 images).

While constitutive expression of *Shi[1]*, even at permissive temperature, with *Rab9*-G4 was lethal, we observed prominently enhanced sleep when *Shi[1]* was conditionally expressed in adulthood using the TARGET system (*McGuire et al., 2004*) (*Figure 3B*). Further, conditional expression of *Shi[1]* with *Rab9*-G4 in the presence of *Repo*-GAL80, an inhibitor of GAL4 localized to all glial cells, prevented the increase in sleep otherwise seen with this driver (*Figure 3C*), concomitant with elimination of barrier expression (*Figure 3—figure supplement 2B*). These data exclude a contribution of neuronal expression of this driver and confirm the sufficiency of the subperineurial glia for the *Shi[1]*-mediated increase in sleep. We confirmed adult-specificity of *Shi[1]* effects by also using the TARGET system to induce *Repo > Shi[1]* expression conditionally during adulthood. A temperature-shift increases sleep in these flies, and upon return to restrictive temperature, the sleep of experimental flies was no longer significantly different from that of controls (*Figure 3—figure supplement 1B*), demonstrating that the phenotype is not likely the result of permanent detrimental effects. Together these data establish that sleep increase upon *Shi* expression is inducible and reversible in adult animals, and is not solely a result of developmental changes.

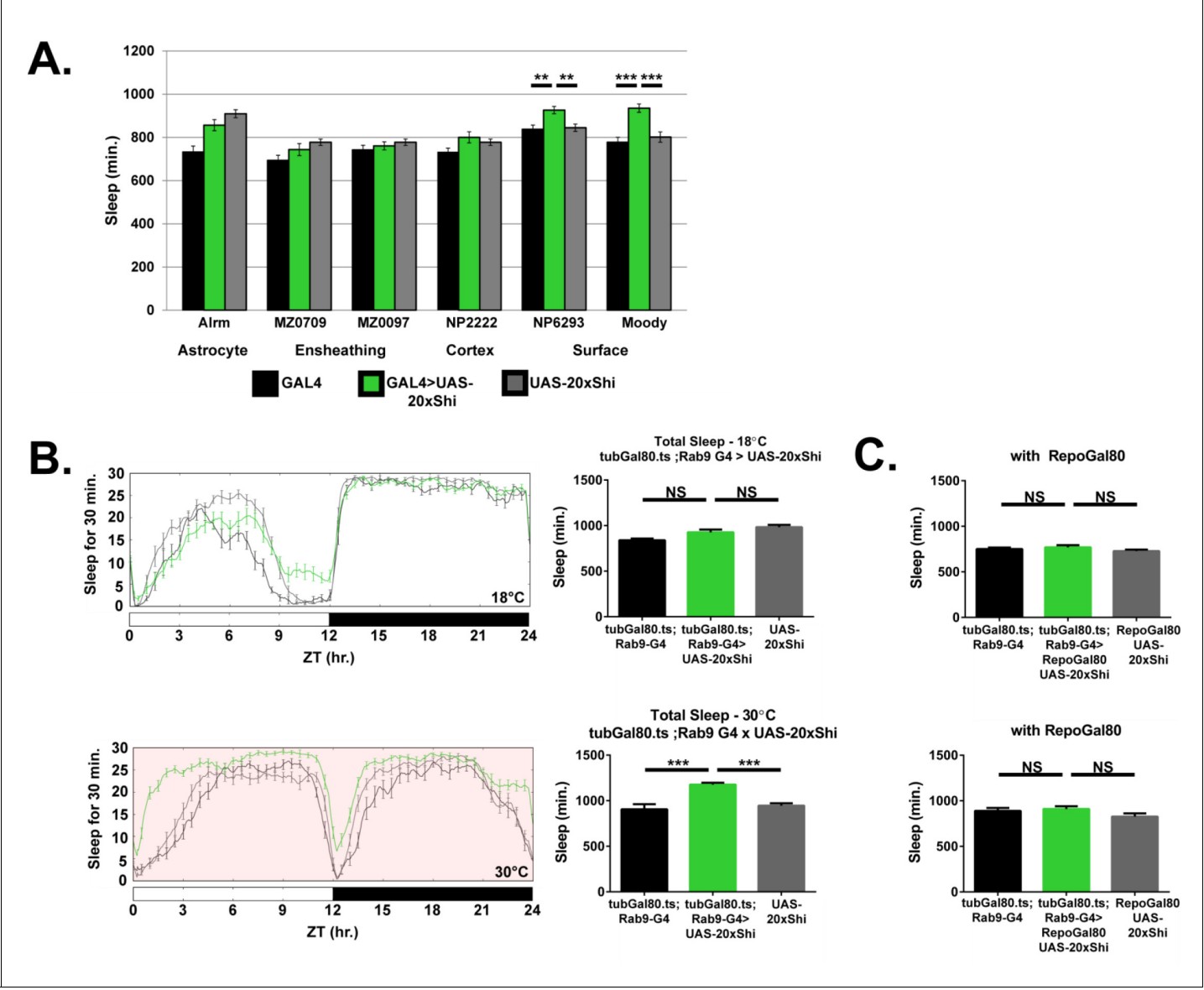

**Figure 3.** Inhibition of endocytosis in surface glia is sufficient to increase sleep. (**A**) Total sleep at 18°C for female flies expressing *Shibire* (UAS-*20xShi. ts1*) by, from left to right, Alrm-GAL4 (n = 16, each genotype), MZ0709-GAL4 (n = 16–32), MZ0097-GAL4 (n = 16–32), NP2222-GAL4 (n = 16–32), NP6293-GAL4 (n = 29–32), *moody*-GAL4 (n = 16). MZ0709, MZ0097, and NP2222 experiments were loaded simultaneously and therefore share the same UAS-Control, graphed repeatedly for each to aid comparison. Significance markings shown for GAL4s in which the experimental group differed significantly from both controls. (**B**) Conditional expression of UAS-*20xShi.ts1* using *tub*Gal80.ts;*Rab9*-GAL4 at permissive (18°C) and restrictive temperature (30°C, red shading) (n = 9–19 females, mean of 4 days). (**C**) Exclusion of glial expression by *Repo*-Gal80 from the conditional expression of UAS-*20xShi.ts1* by *tub*Gal80.ts;*Rab9*-GAL4 at restrictive (18°C) and permissive temperature (30°C) (n = 15–16, mean of 4 days). One-way ANOVA with Holm-Sidak post-hoc test, *p<0.05, **p<0.01, *** is p<0.001. Error bars represent standard error of the mean (SEM).

DOI: https://doi.org/10.7554/eLife.43326.008

The following figure supplements are available for figure 3:

**Figure supplement 1.** Glial adult-specific *Shibire* induction increases sleep and is reversible.
DOI: https://doi.org/10.7554/eLife.43326.009

**Figure supplement 2.** Expression patterns of Rab9-G4 and the BBB drivers.
DOI: https://doi.org/10.7554/eLife.43326.010

**Figure supplement 3.** Perineurial glia are sufficient for *Shibire* rebound phenotype.
DOI: https://doi.org/10.7554/eLife.43326.011

## *Shibire[1]* expression alters ultrastructure but not general integrity of the barrier

Behavioral, electrophysiological or morphological phenotypes have previously been noted from *Shi[1]* expression at permissive temperature (*Gonzalez-Bellido et al., 2009*; *Kilman et al., 2009*). Most prominently, overexpression of *Shi[1]* at 19°C within photoreceptor cells produces a modified ERG profile, accompanied by a decrement in endocytic vesicles and accumulation of microtubule bundles, amounting to gross morphological alteration at the ultrastructural level (*Gonzalez-Bellido et al., 2009*). To determine whether the morphology of fly glia is affected by *Shi[1]* expression, we performed transmission electron microscopy (TEM) on *Repo > Shi[1]* flies raised at permissive temperature. When viewed as a horizontal section through the fly brain, the BBB glia form the two most superficial cell layers which continuously envelope the entire circumference of the brain. The thicker and seemingly less electron-dense PG layer faces the luminal side, and the underlying SPG, considered to be the tight barrier by virtue of septate junctions, appear as a predominantly thinner layer with periodic expanded involutions and protrusions. Examination of the superficial layers reveals a dense and distinct cytoplasmic make-up in *Repo > Shi[1]* brains. Most prominently, amassed and repeated ring-like structures, resembling the microtubule bundles previously reported (*Gonzalez-Bellido et al., 2009*), appear in the experimental animals but are not seen in controls (*Figure 4A*). Additionally, the PG layer appears relatively thinner as compared to controls. This finding shows that in glia, as in neurons, *Shi[1]* expression even at permissive temperature results in a morphological alteration.

The principle role of the hemolymph-brain barrier is to preserve a selective barrier preventing free exchange of hemolymph and brain interstitial fluid. To assess whether expression of *Shi[1]* in glia compromises the general integrity of the barrier, we injected *Repo >Shi[1]* flies with 10kD dextran conjugated to Alexa647, and dissected brains the next day for fixation and analysis by confocal microscopy. In animals with intact barriers, 10kd dextran is restricted from the brain, and forms a layer external to the surface glia (*Pinsonneault et al., 2011*). In both *Repo >Shi[1]* flies as well as parental controls, the fluorescent dextran was seen at the periphery of the brain (*Figure 4B*) indicating that *Shi[1]* expression does not create a leaky barrier. Taken together with the electron micrographs, these images support the finding that with *Shi[1]* expression, general barrier integrity is intact, but intracellular morphology is altered, although the link of the altered morphology to endocytosis is unclear.

## Alterations in specific Rab proteins affect sleep

To better define the relevant pathway, we sought a relatively unbiased method for perturbing trafficking. The Rab proteins are a family of membrane-bound GTPases that demarcate trafficking compartments (e.g. early, recycling, late endosomes, etc) and are known to regulate diverse vesicle movement within the cell, including exo- and endocytotic pathways. We asked which trafficking pathways are important in glia for sleep regulation by screening an existing collection of UAS-Rab lines (*Zhang et al., 2007*), which include both constitutively active (CA) and dominant negative (DN) constructs for the currently described *Drosophila* Rabs. For most Rabs in this collection, two separate insertions exist for each construct (DN and CA), and in the initial screen all available lines were crossed to a pan-glial driver, with hits defined as Rabs that showed opposing effects on sleep of all available CA as compared to all DN lines. In order to avoid potential lethality from overexpression of dominant negative constructs, we used a newly created hormone-inducible pan-glial driver, *Repo*-GeneSwitch (RepoGS). While lethality was not seen for any Rab using this driver, with or without RU486 feeding, we observed that expression was leaky (*Figure 4—figure supplement 1A*), that is not totally dependent on RU486, which has also been reported for other GeneSwitch lines (*Scialo et al., 2016*). Unfortunately, leakiness of RepoGS seemed particularly prominent in the surface glia, and so we examined effects of DN and CA lines on sleep in the presence and absence of RU486. When UAS-Rabs were expressed by RepoGS in the absence of RU, significant differences in sleep between DN and CA lines were found for Rab3, Rab9 and Rab11 (*Figure 4C*). In the presence of RU486, Rab1, Rab5, Rab27 and Rab30 were found significant (*Figure 4—figure supplement 1B*), albeit with only one line available for each DN and CA construct in the case of Rab30.

As preliminary screening compared only experimental flies (RepoGS > UAS Rab) with or without RU, the Rabs of interest were confirmed with *Repo*-GAL4, including the full complement of UAS and

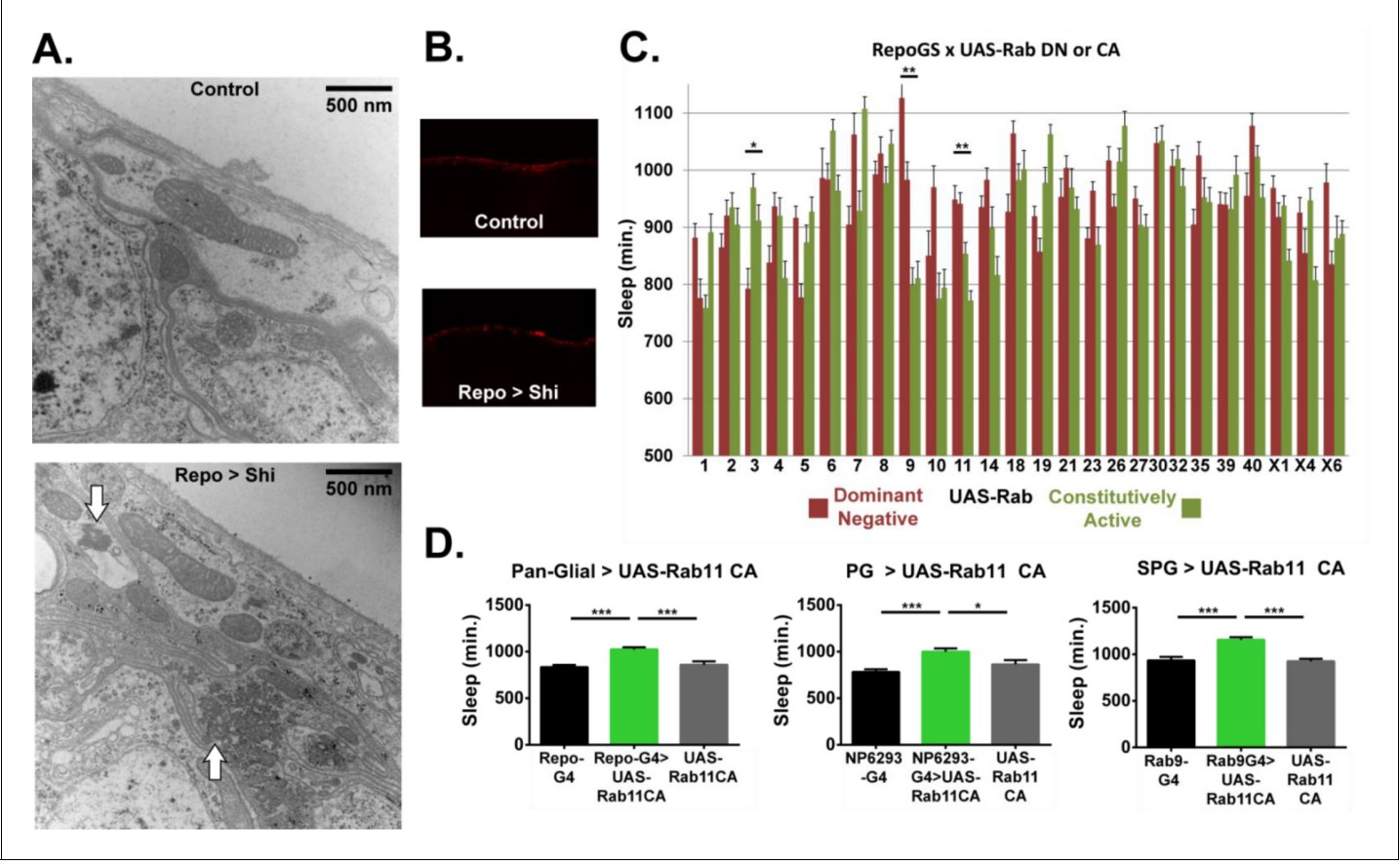

**Figure 4.** Ultrastructural morphology and genetic analysis of Rab proteins support a sleeprelevant function of vesicle trafficking in surface glia. (A) Transmission electron micrographs of surface glia of individual *Repo*-GAL4 > UAS-*20xShi.ts1* and Control female fly brains. Perineurial glia are the most superficial layer with subperineurial glia appearing as the generally thinner and darker layer immediately basal to the perineurial glia. White arrows indicate presence of microtubule bundles. (B) External aspect of hemolymph-brain barrier visualized by injection and fixation of Alexa647-10kd dextran in *Repo*-GAL4 > USA-*20xShi.ts1* and control brains, demonstrating an intact barrier in both genotypes. (C) Total sleep time of RepoGS > UAS Rab CA or DN flies, in the absence of RU486. Red bars represent UAS-Rab DN and green bars are UAS-Rab CA, with two insertions available in most cases (n = 7–16, for all genotypes). Significant Rabs are those in which all DN lines were consistently and significantly different from all CA lines measured by one-way ANOVA with Holm-Sidak post-hoc test or unpaired t-test for Rab30, with *p<0.05, **p<0.01, ***p<0.001. Displayed significance value represents the largest p-value of the 3–4 comparisons. Error bars represent standard error of the mean (SEM). (D) Total sleep time of flies expressing Rab11 CA in all glia (*Repo*-GAL4), perineurial glia (NP6293-GAL4), and subperineurial glia (*Rab9*-GAL4) as compared to GAL4 and UAS controls (n = 15–30), statistics as above.

DOI: https://doi.org/10.7554/eLife.43326.012

The following figure supplements are available for figure 4:

**Figure supplement 1.** Rab screen with pan-glial driver *Repo*-GS and *Repo*-GAL4.
DOI: https://doi.org/10.7554/eLife.43326.013

**Figure supplement 2.** Daytime sleep, sleep deprivation resistance and barrier integrity with Rab11 expression in surface glia.
DOI: https://doi.org/10.7554/eLife.43326.014

GAL4 controls. For Rab3, Rab5 and Rab9, we did not observe sleep increases by this driver (*Figure 4—figure supplement 1C*), and while some Rab27 and Rab30 constructs did exhibit significant increases, we did not find consistent phenotypes when these lines were expressed by surface-glial drivers (data not shown). Rab1 CA also increased sleep relative to parental controls, but given that it affects ER/Golgi transport (*Kiral et al., 2018*), we considered it is less likely to be mechanistically similar to *Shi*[1]. In contrast, expression of recycling-endosome associated *Rab11* CA resulted in a robust sleep increase when driven in all glia (*Figure 4D*) and was significant for both CA lines. Unlike with RepoGS, Rab11 DN driven by *Repo*-GAL4 yielded lethality (*Figure 4—figure supplement 1C*), with no adult flies detected, which may be indicative of *Repo*-GAL4 as a stronger driver. Likewise,

expression of Rab11 DN in either BBB population resulted in lethality (data not shown) while expression of *Rab11* CA in either surface glial population was sufficient to produce increases in total sleep (*Figure 4D*). Unlike with *Shi¹*, we did not see resistance to sleep deprivation with Rab11 driven in glia (*Figure 4—figure supplement 2B*), which could indicate more widespread alteration in trafficking by *Shi¹* than by Rab11 alone. Rab11 is associated with the recycling endosome, and through multiple effectors, is also necessary for endocytic transport, particularly in polarized cells (*Jing and Prekeris, 2009*). Together these results identify the importance of Rab11 in the surface glia, and support the role of endocytic/endosomal trafficking in the sleep function of the surface glia.

## Endocytosis occurs during sleep and is influenced by prior wakefulness

Considering that inhibition of endocytic trafficking in the fly hemolymph-brain barrier increases sleep, could rates of endocytosis at the barrier depend on sleep-wake state? Recent work in mice suggests that sleep serves to promote clearance of interstitial fluid, and consequently harmful metabolites which may have accumulated as a result of sustained neuronal activity during wake, through the glymphatic system (*Xie et al., 2013*). Whether such clearance during sleep is a conserved feature in other organisms is currently unknown, but given that the barrier glia divide brain and body hemolymph in the fly, they are a prime candidate for regulating transport from brain interstitial fluid to the periphery as a function of sleep-wake state.

In order to evaluate effects of sleep-wake state on endocytosis, we collected flies from an early night time point (ZT14 - 2 hr after lights off) and an early day time point (ZT2 - 2 hr after lights on) (*Figure 5A*). At ZT14, sleep pressure is typically high following daytime arousal (flies are diurnal) and a majority of flies experience consolidated sleep, while ZT2 is associated with high locomotor activity. To quantify the amount of endocytosis in the surface glia, we labeled both surface glial layers with UAS-CD8::GFP using the 9–137-GAL4 driver and dissociated the brains to allow access of fluorescence-conjugated 10 kD dextran to the basolateral surface of the barrier (as even a smaller dextran did not penetrate the brain from the apical surface of the BBB *Figure 5—figure supplement 1*). Dissociated brain cells were incubated for 30 min at room temperature, prior to analysis by flow cytometry (Materials and methods) (*Figure 5—figure supplement 2*, for gating strategy). Barrier cells from flies at ZT14 contained significantly higher dextran, suggesting that endocytosis is favored by high sleep or sleep need (*Figure 5A*). To verify that incorporation of dextran into GFP+ surface glial cells reflects endocytosis, samples were also pre-incubated with dynasore (*Macia et al., 2006*), an inhibitor of GTP hydrolysis in dynamin family proteins. At all time points, the AF-dextran signal was significantly diminished by the presence of dynasore (*Figure 5A*), indicating that a substantial portion of the signal is due to endocytosis.

Given that a difference between ZT14 and ZT2 may be indicative of a circadian effect, independent of sleep-wake state, we evaluated flies at an identical time-point (ZT2), but following sleep deprivation (SD). Flies in the SD condition were mechanically sleep deprived for 12 hr beginning at ZT12 and ending at ZT0, and allowed 2 hr to recover, before also being dissected at ZT2. Hence, circadian time is equivalent between SD and non-SD animals, but flies recovering from SD should have greater sleep pressure. We find that flies taken from a state of recovery sleep following deprivation show substantially higher rates of endocytosed AF-dextran than their un-deprived controls at ZT2 (*Figure 5A*) – a level of endocytosis commensurate to that from flies at ZT14.

Brains collected at 6 hr intervals over the course of the 24 hr day showed a rhythmic pattern, which was not significant as a circadian cycle by JTK, but indicated lowest and highest levels of endocytosis at ZT2 and ZT14, respectively (*Figure 5—figure supplement 3B*). To determine when endocytosis declines after the peak at ZT14, we assessed rates over the course of sleep to find that endocytosis is higher at ZT14, as compared to ZT18, ZT22 and ZT2, which are not statistically distinguishable (*Figure 5B*). This would suggest that endocytosis at the barrier is maximal and largely accomplished within the first half of the night. Since enhanced endocytosis at ZT2 following sleep deprivation could be a result of stress rather than enhanced sleep, we fed the sleep-inducing drug, Gaboxadol (GBX) (otherwise known as THIP) (*Berry et al., 2015*; *Dissel et al., 2015*; *Wafford and Ebert, 2006*). While exposure to Gaboxadol during the day did not alter the high endocytosis observed at ZT14 (*Figure 5—figure supplement 3A*), Gaboxadol-treated flies showed increased endocytosis at ZT2 (*Figure 5C*).

Together these findings demonstrate that endocytosis at the hemolymph-brain barrier occurs preferentially during sleep and reflects sleep need. It is normally high during the early night when

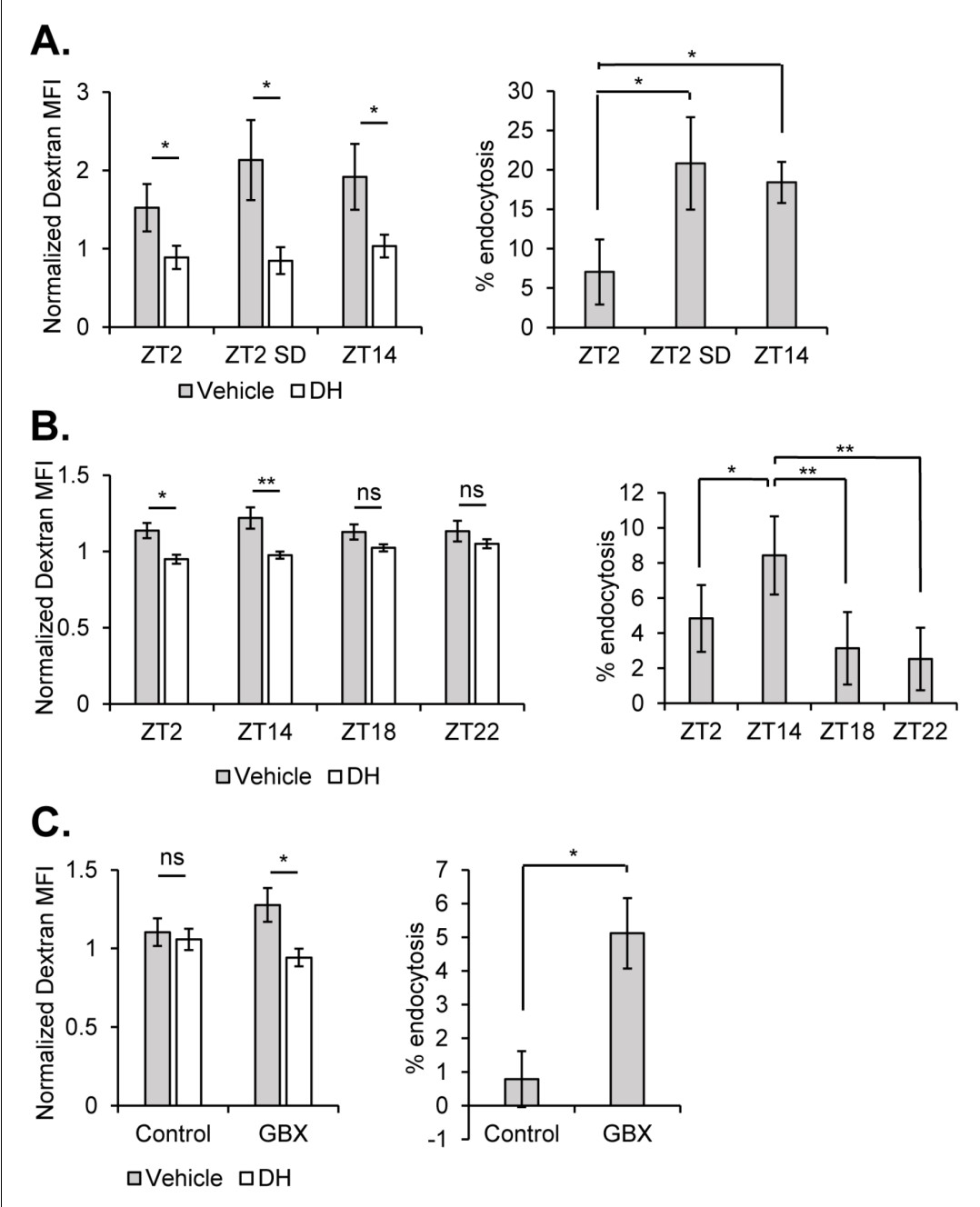

**Figure 5.** Sleep promotes endocytosis at the surface glia. Brains (n = 20, per condition in experiment) from 9-137 GAL4, UAS-CD8::GFP flies were dissected, dissociated, and the samples incubated with Alexa647-conjugated 10kd dextran in the presence or absence of the dynamin inhibitor, dynasore hydrate. Endocytosis measured from surface glial cells by AF647-Dextran signal, expressed as normalized median fluorescence intensity (MFI) (left) and percentage of cells with high endocytosis signal (right). Dextran MFI is normalized to the average MFI of the DH-treated samples. Paired Student's T-test with *p<0.05 and **p<0.01. Percentage of GFP+ cells displaying high signal with vehicle conditions was normalized to inhibitor for each respective timepoint to compare between experiments. One-way ANOVA with correlated measures with post-hoc Tukey's test with *p<0.05 and **p<0.01. (**A**) Endocytosis at ZT2, ZT2 following sleep deprivation (12 hr, mechanical stimulation) and ZT14 (n = 4 per time point, pooled from four experiments). (**B**) Endocytosis at night time points ZT14, ZT18, and ZT22 along with ZT2 (n = 4, pooled from four experiments). (**C**) Endocytosis at ZT2 after feeding of 0.1 mg/mL Gaboxadol or vehicle (n = 3, pooled from two experiments).
DOI: https://doi.org/10.7554/eLife.43326.015

The following figure supplements are available for figure 5:

**Figure supplement 1.** Blood-brain barrier restricts passage of 3kD dextran into the brain.

*Figure 5 continued on next page*

*Figure 5 continued*

DOI: https://doi.org/10.7554/eLife.43326.018

**Figure supplement 2.** Analysis of endocytosis using flow cytometry.

DOI: https://doi.org/10.7554/eLife.43326.016

**Figure supplement 3.** Endocytosis at the BBB across circadian timepoints and upon Gaboxadol feeding at night (ZT14).

DOI: https://doi.org/10.7554/eLife.43326.017

**Figure supplement 4.** Whole-brain biogenic amines and amino acids are unaltered in Repo > Shi flies.

DOI: https://doi.org/10.7554/eLife.43326.019

sleep predominates but can be increased in the early day if flies undergo recovery sleep following deprivation the prior night or if they are induced to sleep by drug treatment.

## Discussion

Glial cells are an important constituent of blood or hemolymph-brain barriers, a conserved feature of complex organisms that protects the central nervous system from unregulated exchange with humoral fluids. In this study, we show that manipulations of endocytosis and vesicular trafficking in the surface glia, particularly the subperineurial glia of the fly BBB, are sufficient to elevate baseline sleep amounts, and that at the BBB, endocytosis itself is influenced by sleep-wake state. Our findings expand the scope of glial involvement in sleep and suggest that trafficking through hemolymph- or blood-brain barriers is an underexplored determinant of sleep-wake behavior.

Expression of *Shibire[1]* has been a popular method to probe circuit-level influence on various fly behavior, with the prevailing interpretation being that exposure to non-permissive temperature abrogates synaptic transmission via an eventual depletion of the synaptic vesicle pool – a consequence of inhibited endocytosis. Ultrastructural work in neurons has revealed, however, that expression of *Shibire[1]* at the lower limit of permissive temperature markedly alters electrophysiological properties and intracellular structures (*Gonzalez-Bellido et al., 2009*), which we also now find in glia. The microtubule defects we report in glia support cellular effects of *Shibire[1]* at permissive temperatures, although they are not necessarily related to behavioral phenotypes of *Shibire[1]* expression in glia (this work) or neurons (*Kilman et al., 2009*) under these conditions. Evidence for dynamin interaction with microtubules has primarily been from early in vitro experiments (*Ferguson and De Camilli, 2012*), with in vivo work showing that only certain dynamin mutations alter microtubules (*Tanabe and Takei, 2009*), underscoring a potentially separable impact of dynamin on endocytosis and microtubules.

*Shibire[1]* expression in *Drosophila* astrocytes was previously shown to disrupt circadian rhythms (*Ng et al., 2011*), but those experiments were designed to prevent expression of *Shibire[1]* at permissive temperatures, leaving open the question of whether other glial phenotypes are independent of temperature. Interestingly, this study showed a decrement in rhythmicity upon expression of *Shibire[1]* at non-permissive temperature in either of the surface glia, although the total percentage of rhythmic animals was unaffected (*Ng et al., 2011*). While sleep was not directly analyzed, a loss of robustness might be explained by increased sleep, present in the context of otherwise intact daily activity patterns. In our experiments, conditional expression of *Shibire[1]* in the surface glia or all glia of adult flies was sufficient to increase sleep, demonstrating that while a phenotype is evident at permissive temperature, it is likewise inducible in adulthood and therefore cannot be accounted for by developmental effects.

Dynamin (*Shibire*) is known to be fundamental to endocytosis and vesicle formation. Nevertheless, the stages of vesicular trafficking are intimately linked, making it difficult to exclusively manipulate, even at the level of molecular machinery, the processes of endocytosis and exocytosis (*Wu et al., 2014*). Indeed, SNARE family members such as synaptobrevin, considered to be specific to exocytosis, have also been found to affect endocytosis (*Xu et al., 2013*; *Zhang et al., 2013*). Therefore, we cannot exclude a contribution of exocytotic processes to the influence of the barrier on sleep, although our data clearly implicate endocytosis in barrier cells.

Rab GTPase family members orchestrate vesicular and membrane traffic within cells including glia (*Dunst et al., 2015*; *Ng and Tang, 2008*) and the BBB, although knowledge of their roles in these populations has been limited. Knockdown of the microtubular motor protein, kinesin heavy chain

(Khc), results in locomotor deficits attributable to developmental alterations in subperineurial glia (*Schmidt et al., 2012*), and also seen with pan-glial Rab21 knockdown. While dynamin also could interact with microtubules, we did not observe locomotor deficits in flies expressing *Shibire[1]*. Through screening of available CA and DN Rab constructs, we identified Rab11 as important for sleep regulation in the barrier glia. We find that Rab11 CA expression, like *Shibire[1]*, produces increases in total sleep when driven in the barrier populations. Constitutive expression of a dominant negative Rab11 is lethal when present in all glia or even just in surface glia, underscoring the importance of this Rab in barrier cells. Rab11 is considered a marker of the recycling endosome compartment, regulating recycling of plasma membrane constituent proteins, although evidence also links Rab11 to more direct participation in early endocytosis. For example, Rab11 functions in *Drosophila* astrocytes to regulate endocytosis of the GABA transporter (GAT) (*Zhang et al., 2017*) and Rab11 knockdown has been shown to inhibit endocytosis and transcytosis of LDL in human iPSC-derived neurons (*Woodruff et al., 2016*). Considering that DN and CA mutations of Rab11 can alter trafficking in the same direction in some conditions (*Khvotchev et al., 2003*), it is possible that Rab11CA inhibits endocytosis at the BBB. Alternatively, Rab11CA could produce a similar phenotype to *Shi[1]* by a different, but related mechanism. For example, while Rab11 DN inhibits recycling of plasma membrane components (*Kiral et al., 2018*), Rab11 CA could enhance recycling. If the mechanism underlying sleep increase at the BBB involves trafficking a particular receptor then according to this model, both inhibiting endocytosis and enhancing recycling would keep this receptor at the membrane.

While we demonstrate that interference with vesicle trafficking and endocytosis at the BBB can alter sleep, and endocytosis in these populations is affected by sleep-wake state, the nature of the sleep-relevant substrates or molecules remains to be determined. The hemolymph-brain barrier, like its mammalian counterpart, contains numerous transporters and receptors (*DeSalvo et al., 2014*) serving to selectively move nutrients and metabolites in, and deleterious compounds such as xenobiotics out. Mutants for the xenobiotic transporter, *Mdr65*, show greater total sleep, but effects have not been directly mapped to the barrier (*Hindle et al., 2017*). We demonstrated recently that efflux transporters in the *Drosophila* BBB are under circadian regulation such that they preferentially pump out xenobiotics during the day. At night, a decrease in transporter activity increases permeability of xenobiotics into the brain (*Zhang et al., 2018*). Based on our current findings, we suggest that activities of the BBB are temporally compartmentalized, such that efflux occurs during the day and endocytosis at night. However, endocytosis is dependent on sleep and sleep history rather than the circadian clock as it will occur during daytime recovery sleep following sleep deprivation at night. Neurotransmitter/modulator levels are known to influence, and vary according to, sleep-wake state (*Sehgal and Mignot, 2011*) and are regulated by glial populations, which could include the BBB glia. To determine if this was the case, we assessed biogenic amine and amino acid content in the brains of *Repo > Shi[1]* flies, but found no consistent differences relative to controls (*Figure 5—figure supplement 4*).

In mammals, a glymphatic system, involving aquaporin channels in astrocytes, promotes flow of interstitial fluid along the brain vasculature and mixing with cerebrospinal fluid to allow exchange of brain fluids and clearance of waste products (*Iliff et al., 2014*; *Xie et al., 2013*). Interestingly, this flow is enhanced during the sleep state (*Xie et al., 2013*). *Drosophila*, as other invertebrates, possess an open circulatory system and thereby lack vasculature within the brain. Nevertheless, separation between the interstitial fluid of the brain and the hemolymph at large is maintained by the hemolymph-brain barrier. Our finding of a role for barrier glia in fly sleep, and the influence of sleep state on endocytosis at the barrier, supports the possibility that clearance or interstitial exchange is a conserved function of sleep. As we find that peripherally injected 10kd dextran does not enter the brain, we speculate that sleep-dependent endocytosis in dissociated BBB cells reflects trafficking from, rather than to, the brain. We suggest that BBB endocytosis occurs during sleep to resolve products of wakefulness and thereby restore metabolic/neural homeostasis, which would account for the increased sleep need that results from a block in endocytosis. In other words, endocytosis would be highest during sleep, and decrease during the night as sleep pressure is resolved. Flies with inhibited endocytosis would be unable to perform this function and resolve sleep pressure, and would in consequence exhibit constantly elevated sleep. By this model, administration of Gaboxadol increases sleep pressure and so is associated with high endocytosis at ZT2 and ZT14, regardless of the baseline levels of endocytosis at these time points.

Sleep was previously linked to permeability of the BBB in rodents, where sleep loss, and even just REM restriction, was shown to increase BBB permeability (*Gómez-González et al., 2013*; *He et al., 2014*; *Hurtado-Alvarado et al., 2016*). These effects were attributed to adenosine (*Hurtado-Alvarado et al., 2016*), which is also considered a somnogenic molecule (*Porkka-Heiskanen and Kalinchuk, 2011*). In fact, adenosine may be relevant to the interaction between astrocytes and sleep (*Fujita et al., 2014*; *Halassa et al., 2009*; *Schmitt et al., 2012*). Given that astrocytes contribute to the mammalian BBB (*Abbott, 2002*), it is possible that sleep effects produced by inhibiting astrocyte signaling (*Halassa et al., 2009*) involve secondary effects on the BBB.

*Drosophila* sleep studies have focused on neuronal circuits controlling sleep-wake (*Artiushin and Sehgal, 2017*; *Dubowy and Sehgal, 2017*), with comparatively few studies investigating the contribution of glia to this circuitry. We identify the barrier glia as regulators of sleep and propose that endocytic trafficking through the barrier is an important function of the sleep state. The specific differences between mammalian and invertebrate barriers notwithstanding, the identification of the glial/endothelial barrier as a population that can mediate changes in daily sleep aligns with emerging ideas of BBB involvement in behavior (*Hoxha et al., 2013*; *Parkhurst et al., 2018*) and sleep function (*Pan and Kastin, 2017*; *Verheggen et al., 2018*), and provides a readily identifiable cellular target to interrogate in other model organisms.

## Materials and methods

### Fly lines

Stocks present in the lab collection include—; ;*Repo*-GAL4/TM3,Sb and UAS-*Shi.ts1*; UAS-*Shi.ts1* (referred to as UAS-*Shi*[1] in the text) and UAS-CD8::GFP and UAS-nGFP. For control genotypes, GAL4 and UAS lines were crossed to iso31.; ;UAS-*20xShi.ts1* (referred to as UAS-*20xShi*[1]) was shared by Gerald Rubin. UAS-*Shi.WT* and UAS-*Shi.K44A* were shared by Konrad Zinsmaier. NP2222-GAL4, NP6293-GAL4, *Alrm*-GAL4, MZ0709-GAL4, MZ0097-GAL4 were shared by Marc Freeman. *Moody*-GAL4 and 9–137 GAL4 was shared by Roland Bainton. UAS-Rab CA and DN lines were acquired from the Bloomington Drosophila Stock Center. *Rab9*-GAL4 (#51587) was also acquired from Bloomington.

To generate the; ;*Repo*-GeneSwitch 2301 line a 4.2 kb genomic fragment (between coordinates 18231884 and 18236068, FlyBase release 6.19) was amplified by PCR and cloned into a PUAST-AttB-Sfi-GeneSwitch vector. Transgenesis for this construct was performed by BestGene at landing site AttP154 (97D2).

### Behavior

Flies were raised in bottles on standard food at either room, or non-permissive temperature (18–19°C) for Shibire and TARGET system (*McGuire et al., 2004*) experiments.

For sleep recording, flies were loaded into glass locomotor tubes containing 5% sucrose in 2% agar, and additionally 0.5 mM RU-486 (mifepristone) in the case of GeneSwitch experiments. Locomotor data was collected using the *Drosophila* Activity Monitoring (DAM) System (Trikinetics, Waltham, MA) and processed using PySolo(*Gilestro and Cirelli, 2009*). Data are processed as 1 min bins, with sleep defined as 5 min without activity. Activity index refers to the average number of beam crossings within an active bout.

Sleep deprivation was performed by mechanical disruption using a vortexer triggered to shake randomly for 2 s out of every 20, for the span of 12 hr beginning at lights-OFF, ZT12. Flies were kept in incubators under 12 hr light: 12 hr dark (LD12:12) cycles and constant temperature (25°C or 18°C or 30°C). In the case of the initial temperature shift experiments with *Shibire*[1], flies were kept at 18°C, and shifted to 30°C for the duration of the night, before being returned to 18°C at lights-ON, ZT0. For behavioral experiments, adult flies of at least 6-days post-eclosion were used, except in the case of tubGal80.ts experiments, where younger flies were loaded to grant additional time to compensate for the loss of strength due to incomplete de-repression of tubGal80.ts.

## Confocal imaging

Fly brains were fixed in 4% PFA in PBS with 0.1% TritonX-100 for 15–20 min, and washed in PBST for 30 min at room temperature, before mounting in VectaShield H-1000 (Vector Laboratories). Confocal imaging was performed on a Leica TCS SP5.

## Electron microscopy

Heads from female *Repo* > UAS-*20xShi*[1] flies approximately two weeks post-eclosion were dissected in a cold room in PBS and fixed with 2.5% glutaraldehyde, 2.0% paraformaldehyde in 0.1M sodium cacodylate buffer, pH7.4, overnight at 4°C. Samples were post-fixed in 2.0% osmium tetroxide for 1 hour at room temperature, and rinsed in $DH_2O$ prior to *en bloc* staining with 2% uranyl acetate. After dehydration through a graded ethanol series, the tissue was infiltrated and embedded in EMbed-812 (Electron Microscopy Sciences, Fort Washington, PA). Flies were not strictly circadian entrained or taken at an exact time of day, but dissections were performed together in the afternoon. Reported images represent sections from single flies from each genotype, although the experimental was compared to both parental controls. The described microtubule structures were not seen in the controls. Embedding, staining and sectioning was performed by the Electron Microscopy Resource Laboratory at Penn. Thin sections were stained with uranyl acetate and lead citrate and examined with a JEOL 1010 electron microscope fitted with a Hamamatsu digital camera and AMT Advantage image capture software.

## Hemolymph-brain barrier permeability

Integrity of the barrier was assessed as previously described (*Pinsonneault et al., 2011*). Female flies were injected with Alexa fluor 647-conjugated 10 kd dextran (ThermoFisher D22914) the day before dissection and kept in standard food vials. Heads were removed and fixed in 4% PFA in PBS for 10–15 min before brains were further dissected out and cleaned. Brains were additionally washed in PBS for 30 min before being mounted in VectaShield H-1000 (Vector Laboratories) and imaged by confocal microscopy.

## HPLC/LC-MS

The brains of female flies ranging from 10 to 20 days post-eclosion were dissected in cold PBS, and collected to 20 brains per vial. Excess PBS was pipetted off after spinning the samples down by micro-centrifuge. The samples were frozen on dry ice and submitted to the Children's Hospital of Pennsylvania Metabolomic Core for HPLC analysis.

## Endocytosis assay and flow cytometry

Flies expressing membrane bound GFP in the surface glia (9–137 GAL4;UAS-CD8::GFP) were dissected in ice-cold AHL (108 mM NaCl, 5 mM KCl, 2 mM $CaCl_2$, 8.2 mM $MgCl_2$, 4 mM $NaHCO_3$, 1 mM $NaH_2PO_4$-$H_2O$, 5 mM trehalose, 10 mM sucrose, 5 mM HEPES; pH 7.5), and 20 brains were collected per condition. To each sample, 60 µL of Collagenase IV (25 mg/mL) and 10 µL of DNase I (1 mg/mL) were added. Following shaking at 37°C for 15 min, brains were broken up by 3 rounds of gently pipetting. Using a 70-µm cell strainer, the brain samples were then transferred to FACS tubes, to which 3 mL of AHL was added and spun down for 5 min (2500 RPM). After removing the supernatant, the pellet was resuspended in AHL and brought to a volume of 200 µL, at which point the sample was split into two new FACS tubes of 100 µL each. One tube received an additional 50 µL of vehicle solution while 50 µL of dynasore hydrate solution (10 µM final concentration) was added to the other. The samples were mixed well and incubated at room temperature for 10 min. Following incubation, 50 µL of Alexa fluor 647-10kd dextran (50 µg/ml final concentration) was added to each tube, mixed, and incubated at room temperature for 30 min. 4 mL of FACS buffer (0.5% BSA w/v + 0.1% w/v sodium azide in PBS) were added to each tube, spun down and the supernatant removed. Samples were immediately analyzed using a FACS Canto II (BD Biosciences).

## Gaboxadol feeding

Gaboxadol hydrochloride (Cayman Chemical Company) was added to standard 5% sucrose in 2% agar food in locomotor tubes at 0.1 mg/mL. For the ZT14 timepoint, flies were flipped to Gaboxadol

food at lights ON (ZT0) while for the ZT2 timepoint they were flipped onto drug just before lights OFF (ZT12). In both cases, they remained on this food for ~14 hr until dissection.

## Live imaging

6-cm plates were coated with poly-L-lysine (0.01%) for greater than 10 min, removed, and let dry. Brains were dissected in AHL, placed on the coated plates, and incubated with 50 µL droplet of 500 µL/mL 3kD FITC-conjugated dextran and either imaged continuously for 10 mins or dye was washed off and brains were imaged in AHL for 5 min. Confocol imaging was performed with 20x submergible water objective (with the lens touching the dye droplet) at excitation/emission wavelengths of 488/520. For a permeable dye, brains were dissected in AHL and incubated with 50 µL droplet of 125 µL/mL Rhodamine B and imaged continuously for 10 min. Imaging was performed with 20x submergible water objective at excitation/emission wavelengths of 543/555.

## Statistical analysis

Graphs and statistical tests were completed using Excel and GraphPad Prism. FACS analysis was performed in FlowJo. For measures of sleep, Controls (GAL4 alone and UAS alone) were compared to Experimental (GAL4 > UAS) animals by one-way ANOVA with Holm-Sidak post-hoc correction. For comparison of endocytosis between ZT2 and ZT14 or ZT2 and ZT2 SD or between each time point control and inhibitor, Students' T test was performed. Additional details regarding tests and significance values are provided in the figure legends.

## Acknowledgements

We thank Zhifeng Yue and Kiet Luu for technical support and members of the Sehgal lab for discussion and reagents. We would like to also thank the Gerald Rubin, Konrad Zinsmaier, Marc Freeman, and Roland Bainton labs for sharing reagents, and the Electron Microscopy Resource Laboratory at Penn and the Children's Hospital of Pennsylvania Metabolomic Core.

## Additional information

### Funding

| Funder | Grant reference number | Author |
|---|---|---|
| National Institutes of Health | T32-HL07953 | Gregory Artiushin |
| National Institutes of Health | R37NS048471 | Amita Sehgal |
| Ellison Medical Foundation | AG-SS-2939-12 | Amita Sehgal |
| Howard Hughes Medical Institute | | Amita Sehgal |

The funders had no role in study design, data collection and interpretation, or the decision to submit the work for publication.

### Author contributions

Gregory Artiushin, Conceptualization, Formal analysis, Investigation, Visualization, Methodology, Writing—original draft, Writing—review and editing; Shirley L Zhang, Formal analysis; Investigation; Visualization; Methodology; Writing—review and editing; Hervé Tricoire, Resources; Writing—review and editing; Amita Sehgal, Conceptualization; Supervision; Funding acquisition; Writing—review and editing

### Author ORCIDs

Shirley L Zhang http://orcid.org/0000-0002-6672-2044
Amita Sehgal http://orcid.org/0000-0001-7354-9641

### Decision letter and Author response

Decision letter https://doi.org/10.7554/eLife.43326.023

Author response https://doi.org/10.7554/eLife.43326.024

## Additional files

### Supplementary files

• Transparent reporting form

DOI: https://doi.org/10.7554/eLife.43326.020

### Data availability

All data generated or analysed during this study are included in the manuscript and supporting files.

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
