## [Decision Letter]

[Editors’ note: a previous version of this study was rejected after peer review, but the authors submitted for reconsideration. The first decision letter after peer review is shown below.]

Thank you for submitting your work entitled "Endocytosis at the blood-brain barrier as a function for sleep" for consideration by *eLife*. Your article has been reviewed by a Senior Editor, a Reviewing Editor, and three reviewers. The following individuals involved in review of your submission have agreed to reveal their identity: Paul Taghert (Reviewer #3).

Our decision has been reached after consultation between the reviewers. Based on these discussions and the individual reviews below, we regret to inform you that your work will not be considered further for publication in *eLife*. However, we do feel the work could be made acceptable but only after the essential changes noted below are handled experimentally. Should you choose to submit a new version of this work, we will endeavor to have it reviewed by the same Board member and referees.

Summary:

This paper explores the functional contributions of the blood brain barrier (BBB) to sleep regulation in *Drosophila*. It uses a full range of genetic cellular and imaging methods to consider the hypothesis that endocytosis/exocytosis in BBB glia changes in ways that supports the role of these cells in helping resolve levels of metabolites that increase in relation to sleep-need. Further it provides evidence for the involvement of glial Rab11 in the cellular mechanism. Experimentally analyzing glial function in sleep, the work touches an aspect of brain physiology that has just begun to be explored but that has been proposed as one of the crucial functions of sleep. Thus, this work is a potentially significant contribution of interest to a general audience.

Though work is clearly described, thorough in most aspects and also several caveats are clearly acknowledged, there are issues that need to be clarified and conclusions that need to be strengthened in order to better establish the key conclusion of the paper that inhibition of endocytosis in glia increases sleep.

Essential revisions:

1) The authors nicely describe various difficulties experienced with genetic tools to perturb endocytosis in targeted glia (e.g. temperature-independent and potentially endocytosis-independent effects of shi-ts expression, as well as leaky GsGal4 expression). In addition, they report similar effects on sleep following expression of constitutively active and dominant negative forms of Rab11 as well as perturbation of Rabs that may not participate in endocytosis (Figure 4 and Figure 4—figure supplement 1). Thus, it is important to better establish sleep phenotypes arise from changes in endocytosis rather than non-specific glial malfunction. Some suggestions of additional experiments to support the hypothesised role of endocytosis are proposed below.

1a) To better characterise sleep-associated endocytosis, it would be useful to measure rates of endocytosis at 4-6 time points to define a daily cycle, not just two time points.

1b) Given the possibility that even adult-specific expression of mutant dynamin or Rab RNAi can have long-lasting changes in glial physiology or alterations in BBB permeability, it would be useful to test if these flies revert to normal sleep behaviour if allowed to recover for an appropriate period of time.

1c) To show that a different amount of sleep deprivation (more or less) produces a proportionate change in endocytosis (i.e., two or more points on the concentration-effect curve).

1d) To show bidirectionality of regulation and to control for potential contribution of stress rather than sleep drive, the authors should test whether endocytosis following more sleep is reduced (this may require orthogonal genetics: e.g. to drive sleep-inducing FB neurons).

2) Different Gal4 lines are used to target subsets of glia. The specificity of these lines needs to be clearly documented. The logic for using one or other for different experiments needs to be clarified. And at least one case, an additional Gal4 may need to be used.

2a) The description of Gal4 expression lines (Figure 3—figure supplement 2) should be expanded Critically, to demonstrate that each one the Gal4s used is indeed expressed preferentially in glia, the figure should include counter stains with glia or neuron-specific antibodies (Repo and Elav, respectively). Glial-specific expression would be also more convincing if the authors could test how well the expression pattern of these Gal4s (Rab9-Gal4 in particular) is blocked by Repo>Gal80. (More minor, it might help if an outline of the brain were drawn).

2b) The tubGal80.ts experiments utilize Rab9-Gal4. To more convincingly implicate the glial cells presumed to be targeted, these experiments should also be repeated with Moody-Gal4 (and possibly Repo-Gal4 as well).

3) Figure 1 shows that of Shi-ts expression in glia induces resistance to mechanical stimulation during sleep. Can a control experiment be done to rule out the possibility of a sensory defect that makes these flies unresponsive to mechanical stimulation? E.g. To test the responsiveness of awake Repo>Shi flies to mechanical stimulation? On this note, are the PG and SPG drivers expressed outside the brain in sensory structures?

Other concerns and suggestions:

4) Is it necessary to dissociate cells before adding Alexa647-dextran? Why not measure before dissociation as most cellular processes including endocytosis could be more robust in an intact brain. If it is simply to allow access to Alexa647-dextran, then could an alternative smaller endocytic tracer be tried?

5) Is resistance to mechanical stimulation following SD specific only to Shi, or also seen following endocytic Rab manipulation?

6) The shi-ts expression results suggest that sleep during SD happens at both temperatures, but nighttime sleep seems to elevated sooner in both males and females at 18 degrees, than in controls. Is this impression correct? and if so, is there an explanation? This difference (if reproducible) involves latencies and time of day differences. The fly sleep literature appears to point to an increasing fragmentation of cellular mechanisms between those that affect daytime versus nighttime sleep. Is there any basis to think or expect that the cell biological mechanisms (exo- and endocytosis) should be geared to daytime versus nighttime?

7) Figure 2C suggests that the effects of Shi-ts on sleep are not due to merely overexpression of Dynamin since the expression of Shi-wt does not have a significant effect. However, for Shi-wt, only total sleep is shown, whereas the main effect for Shi-ts at 18C is on day sleep. To exclude the possibility that a Shi-wt effect on sleep is masked when only total sleep is shown the figure should be revised to include data showing the effect of Shi-wt on daytime sleep.

8) Indeed, it would useful to consistently show data for total sleep and daytime sleep for all genotypes in the paper (these data must be available).

9) In Figure 3A, to exclude possible effects due to expression of Shi in neurons (in addition to expression in the BBB glia), the authors could show that these phenotypes remain when the expression in all glia is blocked using Repo>Gal80.

10) It would be valuable to test, with existing tools, to what extent surface glia are responsible for the phenotypes observed with the pan-glial driver, Repo. Does Shi expression in surface glia also cause resistance to mechanical sleep deprivation? Is the phenotype of either NP6293 or moody >Shi weaker than that of Repo>Shi and does combining NP6293 and moody Gal4s generate a phenotype comparable to that of Repo-Gal4? Why is the 9-137-Gal4 driver (SPG & PG) that is later used in Figure 5 (subsection “Endocytosis occurs during sleep and is influenced by prior wakefulness”) not used here?

11) Figure 3 localizing effect to surface glia: In Figure 3A – which UAS-shi was used? The effect does not appear to match that of repo-Gal4. The difference is no more than 100 min; whereas. in Figure 2A, the difference is >200 min. Is this lack of quantitative matching significant? Maybe it reflects the strength of the Gal4s but alternatively it could mean that other cells contribute. This should be discussed.

12) In the same figure – the lack of effect of astrocyte expression seems to be correlated with an unusually high value for the UAS control. Is there a reason to be concerned here?

13) Regarding EM images in Figure 4A – how abundant and frequent are these ultra-structural changes and how do they correlate with sleep phenotypes?

The observations themselves (amassed ring-like structures resembling microtubules, and a thinner PG layer) are only described to the level of positioning of two arrows in one of the micrographs. There should be some definition of the phenotypes and indication of prevalence, severity or quantification (either within or between animals). Were the suspected MTs coated or bridged as originally described in the paper cited? There no details provided in the methods regarding gender, time of day, age, or "n" value.

In addition, it would be useful to know if these are also seen following induction of wild-type dynamin, because WT-dynamin expression does not cause sleep phenotypes and there is the potential to examine how relevant these ultrastructural defects might be to the sleep effects observed. (Perhaps the cited EM paper has addressed this issue in neurons?) Are there any changes in membranes or vesicular structures that could potentially be connected to a block in endocytosis? What is the predicted EM phenotype associated with Rab11 CA expression?

14) Is the overall structural integrity of BBB, as shown in Figure 4B, also preserved when Rab11CA is expressed in glia?

15) In Figure 4C, the significance is determined based on a difference between DN and CA allele of each Rab protein. Is this the best criterion? Wouldn't it be better to compare each manipulation to the parental controls? In any event, it should be made clear why only Rab11 and not others is deemed to have crossed a significance threshold.

16) The overall logic that explains the link between the manipulations, endocytosis and sleep is confusing. Glial expression of Rab11CA increases sleep, similarly to Shi-ts. Do the authors propose that Rab11CA inhibits endocytosis, similarly to Shi-ts?

17) On similar lines, the data shown in Figure 4C seem to contradict the data in Figure 4D. In Figure 4D, Repo>Rab11CA causes increased sleep, compared to parental controls. Figure 4C, shows that using the RepoGS driver, Rab11 DN induces even more sleep than Rab11CA. Thus, both of these opposing manipulations cause increased sleep. Could it be that both manipulations simply make glial cells sick, and therefore, in both cases, sleep is increased due to malfunction of glia, and not due to specific changes in endocytosis? The possibility for a non-specific effect is supported by similar data in Figure 4—figure supplement 1C, where expression of either DN or CA versions of Rab27 and Rab30 produces exactly the same phenotype of increased sleep, unless the authors believe that these also work by affecting endocytosis. These issues should be explained and addressed in a revised manuscript.

18) In Figure 4—figure supplement 1B – are the effects of Rab3, Rab9 and Rab11 that were significant in Figure 4A (without RU486) still significant in the Figure 4—figure supplement 1B (with RU486)? The significance stars in Figure 4—figure supplement 1B are shown only for Rab1, Rab5, Rab27 and Rab30.

19) The apparent contradiction between Figure 1, Figure 2, Figure 3, Figure 4 and Figure 5 should be more clearly explained. Figure 5 shows that SD (which increases sleep pressure) causes increased endocytosis, but, Figure 1, Figure 2, Figure 3 and Figure 4, show that inhibiting endocytosis actually increases sleep, suggesting that endocytosis in BBB promotes wakefulness. An explanation that connects the two observations should be included.

20) Are the changes in endocytosis that correlate with sleep state specific only to surface glia? Are there changes in endocytosis in other populations of glial cells, as function of sleep?

*Reviewer #1*:

In this manuscript, Artiushin et al., aim to explore the role of glia endocytosis in sleep regulation focussing on a *Drosophila* model of the blood-brain barrier. The work is very interesting in principle as it touches an aspect of brain physiology that has been hardly explored so far but that has been proposed as one of the crucial functions of sleep.

The manuscript is well written in general and the experiments are well reported. The main weakness of the work is that in almost all experiments the genetic tools adopted did not behave as expected. The temperature sensitive form of shibire, (which is instrumental for experiments in Figure 1, Figure 2 and Figure 3) does not appear to provide any specific insight, possibly due to a temperature insensitive activation. The repoGENESwitch tool (instrumental for experiments in Figure 4) does not seem to be properly responsive when it comes to controlling activation.

All in all, the fact that the genetic tools employs are not behaving appropriately is a major weakness for the study as it is basically impossible to understand the exact specificity of the effect in terms of timing. For instance, it is impossible to exclude that this a developmental effect.

One possibility to reinforce the findings could be to adopt different genetic tools with greater reported specificity, such as the ones described in https://pubs.acs.org/doi/10.1021/acssynbio.7b00302

Presentation of data could be improved.

For instance:

# There is no reason to limit labelling of panels within figures. Please use more letters, ideally one per panel. It facilitates discussion.

# Avoid using barplots. Barplots are uninformative and should almost always be discouraged. Use plots that are more informative, giving more insights on the distribution (such as boxplot or violin plots)

# The experiment in Figure 3—figure supplement 2 needs some kind of co-staining confirmation

# Please quantify the observations collected using electron microscopy?

*Reviewer #2*:

Most of the manuscript (Figure 1, Figure 2, Figure 3 and Figure 4) describes how manipulating glia affects sleep. Overall, the behavioral phenotypes are strong. However, there are multiple issues with specificity of the methods, with data representation and with the interpretation of some of the results. Another question is – is it really so surprising and interesting that manipulating large numbers of glial cells has an effect on brain function, and sleep in particular?

The second part (Figure 5) is more interesting and promising, as it provides evidence for correlation between sleep/wake states and the rate of endocytosis in BBB glia. However, this part needs further development. Also, the link between this part and the rest of the manuscript seems a bit artificial, as they do not really support or complement each other well.

Specific comments:

1) The authors show in Figure 1 that expression of the temperature sensitive allele of dynamin, Shi-ts in glia induces resistance to mechanical sleep deprivation. The authors show that the basal wake activity of these flies is not affected. However, another interpretation of the data should be considered. These flies may be unresponsive to mechanical stimulation. The authors should test the responsiveness of awake Repo>Shi flies to mechanical stimulation. The authors should also check if Repo>Shi provides resistance to sleep deprivation induced by alternative methods (thermo-genetic). Finally, the authors test the effects on SD only with Shi, but do not do it with the rest of manipulations, presented later in the paper, such as expressing mutant Rab proteins. Are the effects on resistance to SD specific only to Shi, but not to other manipulations of endocytosis?

2) The authors observe strong effects for the temperature sensitive Shi allele even at the permissive temperature of 18C. This raises the question whether Shi is a good tool in this specific system. The main concern is that the observed phenotypes are mostly due to non-specific effects that make the glial cells sick. The authors should, at least, test if knocking down the endogenous Shi using RNAi produces a phenotype similar to expressing dominant negative Shi.

3) The authors claim in Figure 2C, that the effects of Shi-ts on sleep are not due to merely overexpression of Dynamin since the expression of Shi-wt does not have a significant effect. However, for Shi-wt, only total sleep is shown, whereas the main effect for Shi-ts at 18C is on day sleep. Could it be that the effect of Shi-wt on sleep is masked when only total sleep is shown? The authors should demonstrate the effect of Shi-wt on day sleep.

4) In Figure 3A, to exclude possible effects due to expression of Shi in neurons (in addition to expression in the BBB glia), the authors should show that these phenotypes remain when the expression in all glia is blocked, using Repo>Gal80.

5) To what extent are surface glia responsible for the phenotypes observed with the pan-glial driver, Repo? For instance, does Shi expression in surface glia also cause resistance to mechanical sleep deprivation? Is the phenotype of either NP6293 or moody >Shi weaker than that of Repo>Shi? Will combining both NP6293 and moody Gal4s (SPG & PG) generate a phenotype comparable to that of Repo-Gal4? Why don't the authors use here the 9-137-Gal4 driver (SPG & PG) that is later used in Figure 5 (subsection “Endocytosis occurs during sleep and is influenced by prior wakefulness”)?

6) The images in Figure 3—figure supplement 2 should be of better quality. The outline of the brain should be drawn. The authors also should demonstrate that each one the Gal4s used in this Figure is indeed expressed preferentially in glia by co-staining with glia or neuron-specific antibodies (Repo and Elav, respectively). The glia-specific expression would be also more convincing if the authors could show the expression pattern of these Gal4s (Rab9-Gal4 in particular) when the expression of the UAS-nGFP reporter is blocked in glia, using Repo>Gal80.

7) The EM images in Figure 4A – how abundant and frequent are these ultra-structural changes? Are these changes also observed upon the expression of Rab11CA? Also, could the authors detect any accumulation of vesicles that cannot be secreted due to expression of mutant Shi?

8) Is the overall structural integrity of BBB, as shown in Figure 4B, also preserved when Rab11CA is expressed in glia?

9) In Figure 4C, the significance is determined based on a difference between DN and CA allele of each Rab protein. Is this the best criterion? Wouldn't it be better to compare each manipulation to the parental controls? Also, what would be the effect of overexpressing WT versions of these Rab proteins?

10) The overall logic that explains the link between the manipulations, endocytosis and sleep is confusing. Glial expression of Rab11CA increases sleep, similarly to Shi-ts. Do the authors propose that Rab11CA inhibits endocytosis, similarly to Shi-ts?

Do they suggest that expression of constitutively active Rab11 is equivalent to the inhibition of vesicular trafficking? If so, what would be the effect of Rab11 RNAi? Would it produce an opposite phenotype (less sleep)? Repo>Rab11 DN could be useful, but it was lethal. The authors should try using tub-gal80ts, to achieve inducible expression of Rab11 DN. Also, does Repo>Rab11CA cause resistance to mechanical sleep deprivation, just like Shi-ts?

11) One of the major claims of this paper is that inhibition of endocytosis in glia (Shi-ts, Rab11CA) increases sleep. It is important to show that an opposite manipulation of increasing endocytosis actually reduces sleep. The data shown in Figure 4C is confusing and seems to contradict the data in Figure 4D. In Figure 4D, Repo>Rab11CA causes increased sleep, compared to parental controls. The interpretation is that this happens due to inhibition of endocytosis. However, in Figure 4C, using RepoGS driver, Rab11 DN induces even more sleep than Rab11CA. It seems that both of these opposing manipulations cause increased sleep, which is counter-intuitive. Could it be that both manipulations simply make glial cells sick, and therefore, in both cases, sleep is increased due to malfunction of glia, and not due to specific changes in endocytosis? The possibility for a non-specific effect is supported by similar data in Figure 4—figure supplement 1C, where expression of either DN or CA versions of Rab27 and Rab30 produces exactly the same phenotype of increased sleep.

12) In Figure 4—figure supplement 1B – are the effects of Rab3, Rab9 and Rab11 that were significant in Figure 4A (without RU486) still significant in the Figure 4—figure supplement 1B (with RU486)? The significance stars in Figure 4—figure supplement 1B are shown only for Rab1, Rab5, Rab27 and Rab30.

13) How do the authors explain the phenotypes caused by the expression of other Rabs, besides Rab11? Do they think that they all work by affecting endocytosis?

14) In Figure 5, the authors should demonstrate that their genetic manipulations (Shi-ts and Rab11CA) actually can inhibit endocytosis in the surface glia using the 10kd dextran absorption assay. Also, could it be that the changes they see after SD are in fact caused by physical damage to the BBB? They may test if the effects of SD on endocytosis are resolved once the flies are allowed few hours of recovery after SD. They can also test how thermo-genetic SD affects their assay. They should also do EM on BBB after SD.

15) There is an apparent contradiction between Figure 1, Figure 2, Figure 3Figure 4 and Figure 5. According to Figure 5, SD causes increased endocytosis. SD of course increases sleep pressure. But, in the Figure 1, Figure 2, Figure 3 and Figure 4Figures, the authors claim that inhibiting endocytosis actually increases sleep, suggesting that endocytosis in BBB promotes wakefulness. A possible solution to this paradox is that during normal sleep, there is an increase in endocytosis. This is important to support the function of sleep because increased endocytosis eventually helps to reduce sleep pressure. This is why inhibition of endocytosis promotes sleep – because in the absence of proper endocytosis, sleep quality is compromised, and sleep is then less effective in reducing the sleep pressure.

16) Are the changes in endocytosis that correlate with sleep state specific only to surface glia? The authors should have additional control in Figure 5 – test if there are any changes in endocytosis in other populations of glial cells, as function of sleep.

*Reviewer #3*:

This paper explores the functional contributions of the blood brain barrier (BBB) to sleep regulation in the model system *Drosophila*. It is a careful study that uses a full range of genetic cellular and imaging methods to consider the hypothesis that endocytosis/exocytosis in BBB glia changes in ways that supports the role of these cells in helping resolve levels of metabolites that increase in relation to sleep-need. Further it makes a strong case for the involvement of Rab11 in the cellular mechanism. The emerging involvement of glial compartments in the fundamentals of brain physiology (here sleep regulation) makes this work a significant contribution of interest to a general audience. The work is clearly described, thorough in most aspects of the analysis and discussed in a scholarly fashion. I have a few questions and concerns that may help improve the paper.

Shibere mis-expression and effects on sleep.

The paper wisely pays careful attention to the effects (and lack of effects) of temperature with a classic ts allele of dynamin. In addition, I also commend the use of different shi isoforms to help focus on the biologically (not technically) important results. I had one question about this dataset that involved, not amount of sleep, but its temporal patterning.

The results suggest that sleep during SD happens at both temperatures but I notice that nighttime sleep is elevated sooner in both males and females at 18 degrees, than in controls. Is this impression correct? and if so, is there an explanation? This difference (if reproducible) involves latencies and time of day differences. In my reading of the fly sleep literature, there is an increasing fragmentation of cellular mechanisms between those that affect daytime versus nighttime sleep. Is there any basis to think or expect that the cell biological mechanisms (exo- and endocytosis) should be geared to daytime versus nighttime?

Dynamin and MTs

The ultrastructural work is an effective supporting body of evidence to evaluate the effects of UAS-shi at permissive temperatures. However, the results – amassed ring-like structures resembling microtubules, and a thinner PG layer – were not well-described, beyond the positioning of two arrows in one of the micrographs. There was no clear definition of the phenotypes nor any indication of prevalence, severity or quantification (either within or between animals). Were the suspected MTs coated or bridged as originally described in the paper cited? There no details provided in the methods regarding gender, time of day, age, or "n" value. On a tangential (but relevant) note, was ultrastructure observed at only a single phase point? This issue is clearly beyond the scope of the current study.

Cell Specificity

Figure 3 localizing effect to surface glia;

3A – which UAS-shi was used?

- Male or female flies?

- The effect does not appear to match that of repo-Gal4. Difference is no more than 100 m; whereas. In Figure 2A, the difference is >200 min. Is this lack of quantitative matching significant? Maybe it reflects the strength of the Gal4s but alternatively it could mean that other cells contribute. I did not find any discussion of this point.

Rab involvement

The screens were careful, unbiased, quantitative and very large in scope. I commend the authors for determining that the Gs-Gal4 line was leaky but was somewhat confused by the overall definitions of hits. There was a set of hits with no RU486 (Rab 3, 5 and 9) and a different one with RU486 (Rab1, 5, 27 and 30). Yet at the end, only Rab11 is judged to have experimental support – I was not clear why (1) only Rab11 and not others crossed threshold. Also (2) whether Rab11 is a positive or negative regulator of sleep.

“As preliminary screening compared only experimental flies (RepoGS>UAS-Rab) with or without RU, the Rabs of interest were confirmed with Repo-GAL4, including the full complement of UAS and GAL4 controls.”

1) But only Rab 11 is shown, not Rab 3 or Rab9 – follow-up on Rabs 3 and 9 were neither shown nor mentioned,

“Expression of recycling-endosome associated Rab11 CA resulted in a robust sleep increase when driven in all glia (Figure 4D) and was significant for both CA lines.”

2) Paradoxically, Rab11 activity appears to increase sleep in the follow-up but in the preliminary large scale screen the DN isoforms of Rab11 produced significantly more sleep than did the CA isoforms – by my reading that means Rab11 in glia is a negative regulator. So, I'm not sure how to interpret the combined dataset.

Endocytosis

1) Why dissociate cells before adding Alexa647-dextran? Why not measure before dissociation? The results are fortunately very clear, but I would think cellular processes like endocytosis would be more robust in an intact brain.

2) Also it would be a stronger case if a different amount of SD produces a proportionate change in endocytosis (i.e., two points on the concentration-effect curve).

Also, would it would strengthen the case to measure endocytosis following more sleep (to control for potential contribution of stress to change); This would require orthogonal genetics to drive sleep-inducing FB neurons, so I suspect the genetics would be too complicated for a two-month period of additional study.

---

## [Author Response]

[Editors’ note: the author responses to the first round of peer review follow.]

This paper explores the functional contributions of the blood brain barrier (BBB) to sleep regulation in Drosophila. It uses a full range of genetic cellular and imaging methods to consider the hypothesis that endocytosis/exocytosis in BBB glia changes in ways that supports the role of these cells in helping resolve levels of metabolites that increase in relation to sleep-need. Further it provides evidence for the involvement of glial Rab11 in the cellular mechanism. Experimentally analyzing glial function in sleep, the work touches an aspect of brain physiology that has just begun to be explored but that has been proposed as one of the crucial functions of sleep. Thus, this work is a potentially significant contribution of interest to a general audience.Though work is clearly described, thorough in most aspects and also several caveats are clearly acknowledged, there are issues that need to be clarified and conclusions that need to be strengthened in order to better establish the key conclusion of the paper that inhibition of endocytosis in glia increases sleep.Essential revisions:1) The authors nicely describe various difficulties experienced with genetic tools to perturb endocytosis in targeted glia (e.g. temperature-independent and potentially endocytosis-independent effects of shi-ts expression, as well as leaky GsGal4 expression). In addition, they report similar effects on sleep following expression of constitutively active and dominant negative forms of Rab11 as well as perturbation of Rabs that may not participate in endocytosis (Figure 4 and Figure 4—figure supplement 1). Thus, it is important to better establish sleep phenotypes arise from changes in endocytosis rather than non-specific glial malfunction. Some suggestions of additional experiments to support the hypothesised role of endocytosis are proposed below.1a) To better characterise sleep-associated endocytosis, it would be useful to measure rates of endocytosis at 4-6 time points to define a daily cycle, not just two time points.

We have expanded the nighttime time points and included these findings in (Figure 5B). We also show time points around the clock (Figure 5—figure supplement 2B).

1b) Given the possibility that even adult-specific expression of mutant dynamin or Rab RNAi can have long-lasting changes in glial physiology or alterations in BBB permeability, it would be useful to test if these flies revert to normal sleep behaviour if allowed to recover for an appropriate period of time.

To examine whether adult-induced expression of Shi produces irreversible alterations in sleep, which could signify permanent changes in function of the barrier, we expressed Shi in glia under the control of tubGal80.ts. In these experiments, temperature-dependent expression of tubGal80.ts suppresses Shi expression during development but allows expression in adults. We find that sleep is increased upon adult expression of Shibire and returns to levels which are not significantly different from controls when expression is blocked (in each case with a temperature shift) (Figure 3—figure supplement 1B).

1c) To show that a different amount of sleep deprivation (more or less) produces a proportionate change in endocytosis (i.e., two or more points on the concentration-effect curve).

Although this is a good point and we would certainly like to resolve this question, there are a number of scientific and technical difficulties that prevent us from doing so. First, more or less deprivation does not necessarily correlate with a linearly proportionate response, whether in behavioral rebound or cellular measures (for instance, 12 hours of deprivation does not necessarily produce a larger rebound than 6 hours). Second, due to the degree of variability in the endocytosis assay, we do not have the resolution to detect such a difference. If we expand our time points after a period of deprivation, we still may not be positioned to detect a difference. We have examined later night time points (Figure 5B) and found that while endocytosis is high during early night sleep and low by the end of the night/early day, it is difficult to determine a difference in the time points in between, meaning that endocytosis may be largely complete within the early hours of sleep or we simply do not have the resolution with this technique.

1d) To show bidirectionality of regulation and to control for potential contribution of stress rather than sleep drive, the authors should test whether endocytosis following more sleep is reduced (this may require orthogonal genetics: e.g. to drive sleep-inducing FB neurons).

To examine potential bidirectionality and assess the contribution of stress, we have elected to use Gaboxadol to induce sleep rather than introducing the added complications of orthogonal genetics and the requisite temperature-induction. We found that enhancing sleep throughout the day did not decrease endocytosis at ZT14, but endocytosis was increased at ZT2 if flies were treated with Gaboxadol prior to this point (Figure 5C). We suggest that Gaboxadol increases sleep pressure and so promotes endocytosis at all times, thereby accounting for lack of a decrease at ZT14. Importantly, these data indicate that it is not the stress of sleep deprivation which is responsible for elevating endocytosis, as two methods of inducing sleep at a time when wake predominates both increase endocytosis.

2) Different Gal4 lines are used to target subsets of glia. The specificity of these lines needs to be clearly documented. The logic for using one or other for different experiments needs to be clarified. And at least one case, an additional Gal4 may need to be used.

In addition to the discussion of sub-points below, we have added a few remarks throughout the text clarifying why the given Gal4 lines were used for particular experiments.

2a) The description of Gal4 expression lines (Figure 3—figure supplement 2) should be expanded Critically, to demonstrate that each one the Gal4s used is indeed expressed preferentially in glia, the figure should include counter stains with glia or neuron-specific antibodies (Repo and Elav, respectively). Glial-specific expression would be also more convincing if the authors could test how well the expression pattern of these Gal4s (Rab9-Gal4 in particular) is blocked by Repo>Gal80. (More minor, it might help if an outline of the brain were drawn).

The Gal4 lines used in this study, other than Rab9-G4, have all been previously described and used for glial expression. We have emphasized this point and ensured that appropriate references for all drivers are included in the text.

The Gal4 lines we used (Figure 3—figure supplement 2) have limited expression, if any, outside the BBB; given that, it would be difficult to co-localize expression with a pan-neuronal or pan-glial stain. In lieu of a counter-stain, we have provided Gal4>GFP expression in the presence and absence of *repo*-Gal80, which by comparison makes clear what expression is neuronal and what glial (Figure 3—figure supplement 2B). This is particularly evident for the drivers relevant for behavioral differences, such as Rab9-Gal4, as the surface glia are easily distinguishable from all other cell types. As requested by the reviewer, we provide an outline of the brain in Figure 3—figure supplement 2A.

2b) The tubGal80.ts experiments utilize Rab9-Gal4. To more convincingly implicate the glial cells presumed to be targeted, these experiments should also be repeated with Moody-Gal4 (and possibly Repo-Gal4 as well).

The SPG glial cells are implicated by two drivers, Rab9-Gal4 and Moody-Gal4. To confirm the adult-specific effect of Shi, we now provide the tubGal80.ts experiment with a second driver, Repo-Gal4 (Figure 3—figure supplement 1). The data confirm that conditional adult expression is sufficient for the glial sleep phenotype.

3) Figure 1 shows that of Shi-ts expression in glia induces resistance to mechanical stimulation during sleep. Can a control experiment be done to rule out the possibility of a sensory defect that makes these flies unresponsive to mechanical stimulation? E.g. To test the responsiveness of awake Repo>Shi flies to mechanical stimulation? On this note, are the PG and SPG drivers expressed outside the brain in sensory structures?

To rule out the possibility that a sensory defect in Repo>Shi flies makes them unresponsive to the mechanical stimulation of sleep deprivation, we stimulated the experimental and control flies at ZT11 – 12, a time when flies are predominantly awake, as suggested. We find no difference in the resultant beam crossings induced by stimulation between Repo>Shi flies and controls (Figure 1—figure supplement 1), suggesting that resistance to sleep deprivation cannot be accounted for by impaired sensitivity to the stimulus. Also, we are not aware of sensory structure expression of PG and SPG.

Other concerns and suggestions:4) Is it necessary to dissociate cells before adding Alexa647-dextran? Why not measure before dissociation as most cellular processes including endocytosis could be more robust in an intact brain. If it is simply to allow access to Alexa647-dextran, then could an alternative smaller endocytic tracer be tried?

While we agree that a minimally perturbed sample might better reflect cellular processes occurring in vivo, in our hands we have found that dissociation is necessary to allow access of AF647-Dextran to both sides of the barrier. Even with a smaller (3KD) dextran we did not observe any appreciable penetration into an intact brain (Figure 5—figure supplement 3).

5) Is resistance to mechanical stimulation following SD specific only to Shi, or also seen following endocytic Rab manipulation?

We have added this experiment (Figure 4—figure supplement 2). Resistance to mechanical stimulation appears to be specific to Shi, with surface glial being sufficient. It is plausible that Shi provides a more general perturbation of trafficking than Rab11, hence capturing both baseline and resistance phenotypes. These data also demonstrate that these effects can be separated as Rab11 only affects baseline sleep.

6) The shi-ts expression results suggest that sleep during SD happens at both temperatures, but nighttime sleep seems to elevated sooner in both males and females at 18 degrees, than in controls. Is this impression correct? and if so, is there an explanation? This difference (if reproducible) involves latencies and time of day differences. The fly sleep literature appears to point to an increasing fragmentation of cellular mechanisms between those that affect daytime versus nighttime sleep. Is there any basis to think or expect that the cell biological mechanisms (exo- and endocytosis) should be geared to daytime versus nighttime?

We are not sure that this difference in latencies that the reviewer mentions is a consistent finding in our data. The difference in latency between 18 and 30 degrees could also be a consequence of the shift in sleep patterns related to temperature, as touched on elsewhere in the text.

With respect to time-of-day differences, it is true that some publications demonstrate effects predominantly on daytime or nighttime sleep when given neuronal populations are manipulated, but in our view, there is not yet a clear delineation between circuits and mechanisms which govern daytime vs. nighttime sleep. Likewise, there is little evidence to suggest that daytime sleep and nighttime sleep serve different functions in the fly.

7) Figure 2C suggests that the effects of Shi-ts on sleep are not due to merely overexpression of Dynamin since the expression of Shi-wt does not have a significant effect. However, for Shi-wt, only total sleep is shown, whereas the main effect for Shi-ts at 18C is on day sleep. To exclude the possibility that a Shi-wt effect on sleep is masked when only total sleep is shown the figure should be revised to include data showing the effect of Shi-wt on daytime sleep.

The figure has been updated to include daytime sleep. The result is still the same – Shi-wt is not significantly different from both controls, in terms of total or day sleep.

8) Indeed, it would useful to consistently show data for total sleep and daytime sleep for all genotypes in the paper (these data must be available).

Daytime and total sleep figures have now been included, either in the main text or supplements in order to not crowd the figures.

9) In Figure 3A, to exclude possible effects due to expression of Shi in neurons (in addition to expression in the BBB glia), the authors could show that these phenotypes remain when the expression in all glia is blocked using Repo>Gal80.

We showed that the increased sleep phenotype is present with Repo-G4 (only glial expression) as well as Rab9-G4 (BBB expression), but not when glial expression of Rab9-Gal4 is blocked. Comparing expression of surface glial drivers (moody, Rab9, NP6293) in the presence and absence of *repo*-Gal80 (Figure 3—figure supplement 2B) shows little or, in the case of moody-G4, no expression in neurons, therefore making it quite unlikely that the phenotype is due to non-glial expression.

10) It would be valuable to test, with existing tools, to what extent surface glia are responsible for the phenotypes observed with the pan-glial driver, Repo. Does Shi expression in surface glia also cause resistance to mechanical sleep deprivation? Is the phenotype of either NP6293 or moody >Shi weaker than that of Repo>Shi and does combining NP6293 and moody Gal4s generate a phenotype comparable to that of Repo-Gal4? Why is the 9-137-Gal4 driver (SPG & PG) that is later used in Figure 5 (subsection “Endocytosis occurs during sleep and is influenced by prior wakefulness”) not used here?

We conducted additional experiments to address this question and find that the resistance to sleep deprivation phenotype indeed tracks to surface glia, with NP6293 G4 being sufficient (Figure 3—figure supplement 3).

In the screen of glial sub-type drivers, we chose to use the individual surface glial drivers instead of 9-137 G4 in order to establish the effects of each population in isolation, just as the other glial drivers were chosen for specificity to individual glial sub-types. Nevertheless, Shi expression with 9-137 G4 also produces an increase in sleep, albeit smaller than would be expected if combining the effects of the surface glial lines individually (and smaller than that produced by repo-Gal4). This may be a consequence of differences in driver strength, which are not readily assessable or comparable. Given that NP6293 G4 and moody G4 are on the same chromosome, the effort required to perform these experiments may not be worth the potential result because there still may be a discrepancy in driver strength between the surface glial drivers and repo.

11) Figure 3 localizing effect to surface glia: In Figure 3A – which UAS-shi was used? The effect does not appear to match that of repo-Gal4. The difference is no more than 100 min; whereas. in Figure 2A, the difference is >200 min. Is this lack of quantitative matching significant? Maybe it reflects the strength of the Gal4s but alternatively it could mean that other cells contribute. This should be discussed.

A line has been added regarding this point. As the reviewers point out, it may very well be a difference in driver strength, and/or that repo-G4 encompasses both surface glial populations while the lines in Figure 3A do not.

12) In the same figure – the lack of effect of astrocyte expression seems to be correlated with an unusually high value for the UAS control. Is there a reason to be concerned here?

We do not believe so as the UAS control does not achieve ceiling levels of sleep, so there is still room for an increase. Also, as sleep varies some from experiment to experiment, we are careful to always compare to controls within the same experiment. Importantly, a different astrocyte driver also does not increase sleep significantly above the controls. As these data used the other Shibire line, rather than the 20xShibire used in much of the manuscript and in the rest of Figure 3A, we did not include them in the manuscript, but provide them for the reviewer (Author response image 1).

**Author response image 1. respfig1:** Shibire expression in astrocyte-like glia does not increase sleep. Total and daytime sleep of *Eaat1*-GAL4>UAS*-Shi.ts1*;UAS-*Shi.ts1* female flies at 18°C (n=16, each genotype). One-way ANOVA with Holm-Sidak post-hoc test, * P < 0.05, ** P < 0.01, *** P < 0.001. Error bars represent standard error of the mean (SEM).

13) Regarding EM images in Figure 4A – how abundant and frequent are these ultra-structural changes and how do they correlate with sleep phenotypes?The observations themselves (amassed ring-like structures resembling microtubules, and a thinner PG layer) are only described to the level of positioning of two arrows in one of the micrographs. There should be some definition of the phenotypes and indication of prevalence, severity or quantification (either within or between animals). Were the suspected MTs coated or bridged as originally described in the paper cited? There no details provided in the methods regarding gender, time of day, age, or "n" value.

The details requested regarding the animals used have now been provided in the methods and/or figure caption. Briefly, our intention is to highlight the presence of unusual MT and intracellular structure in glial cells expressing Shi at 18 degrees. This simply demonstrates that by EM, expression of Shi at permissive temperature is not without consequence in glia, as has also been shown for neurons in the paper cited. While perhaps we could quantify something like the number of MTs over a length of surface glial area, it is not clear that this or any other quantification will be a useful measurement, because the phenotype is binary – the structures are present in Repo>Shi tissue, and not found in controls.

In addition, it would be useful to know if these are also seen following induction of wild-type dynamin, because WT-dynamin expression does not cause sleep phenotypes and there is the potential to examine how relevant these ultrastructural defects might be to the sleep effects observed. (Perhaps the cited EM paper has addressed this issue in neurons?) Are there any changes in membranes or vesicular structures that could potentially be connected to a block in endocytosis? What is the predicted EM phenotype associated with Rab11CA expression?

Our hope was to identify membrane or vesicular structures in the control surface glial images to have a target for comparison with the experimental images. Unfortunately, perhaps due to resolution or preparation, we were unable to pinpoint obvious endocytic or vesicular trafficking events in the control images, thereby making it hard to compare to Shi flies and to say, beyond the gross changes described, where the alteration connected to blocking endocytosis is expressed.

Determining whether WT-dynamin causes ultrastructural defects could be interesting, but even if so, we could not exclude a contribution of such defects to the sleep phenotype produced by Shibire as these defects may be necessary but not sufficient.

14) Is the overall structural integrity of BBB, as shown in Figure 4B, also preserved when Rab11CA is expressed in glia?

Yes, the barrier does not appear permissive to dextran when Rab11CA is expressed (Figure 4—figure supplement 2).

15) In Figure 4C, the significance is determined based on a difference between DN and CA allele of each Rab protein. Is this the best criterion? Wouldn't it be better to compare each manipulation to the parental controls? In any event, it should be made clear why only Rab11 and not others is deemed to have crossed a significance threshold.

We agree that the ideal comparison would be between each manipulation and its respective parental controls. But since our initial purpose was to screen as many Rabs as possible (which included multiple CA and DN lines per Rab in most cases), we chose to compare CA and DN effects of the experimental animals only. As noted below, we expected that CA and DN would have opposing effects, although this is not always the case. We followed up Rabs that were promising in the initial screen by using Repo-G4 and including all of the parental controls. Rab11 was a positive in the DN/CA screen and, in subsequent assays, Rab11 CA consistently increased sleep relative to controls, regardless of whether it was expressed with Repo or BBB drivers. This is now clarified in the text.

16) The overall logic that explains the link between the manipulations, endocytosis and sleep is confusing. Glial expression of Rab11CA increases sleep, similarly to Shi-ts. Do the authors propose that Rab11CA inhibits endocytosis, similarly to Shi-ts?

Rab11 can affect endocytosis, as knockdown of Rab11 has been shown to inhibit endocytosis and transcytosis, for example, of LDL (Woodruff et al., 2016). Assuming that CA and DN have opposing effects, Rab11CA would not be expected to block endocytosis; however, DN and CA versions can sometimes go in the same direction, with WT or CA mutants blocking the pathway at a specific point due to limiting amounts of binding partners. In addition, for the Rab collection we used, the Rab CA and DNs are called so based on the best prediction that the sites mutated will be GTPase-defective and GTP-binding defective, rather than functional confirmation of their CA and DN activity; in fact, Rab3 is mislocalized with CA and DN mutations (Zhang et al., 2007).In the case of Rab11, in particular, both the DN and CA mutations we used alter trafficking in one direction in certain contexts (Khvotchev et al., 2003).

Therefore, it is possible that Rab11 CA inhibits endocytosis. Alternatively, if the mechanism underlying the sleep phenotype involves, for example, a certain receptor, it is also possible that Rab11 CA and Shi lead to the same behavioral phenotype by keeping the receptor on the plasma membrane – Rab11CA by promoting its recycling, and Shi by inhibiting its endocytosis.

17) On similar lines, the data shown in Figure 4C seem to contradict the data in Figure 4D. In Figure 4D, Repo>Rab11CA causes increased sleep, compared to parental controls. Figure 4C, shows that using the RepoGS driver, Rab11 DN induces even more sleep than Rab11CA. Thus, both of these opposing manipulations cause increased sleep. Could it be that both manipulations simply make glial cells sick, and therefore, in both cases, sleep is increased due to malfunction of glia, and not due to specific changes in endocytosis? The possibility for a non-specific effect is supported by similar data in Figure 4—figure supplement 1C, where expression of either DN or CA versions of Rab27 and Rab30 produces exactly the same phenotype of increased sleep, unless the authors believe that these also work by affecting endocytosis. These issues should be explained and addressed in a revised manuscript.

We have driven Rab27 and Rab30 with the surface glial drivers, but have not found consistent phenotypes when limiting expression to these glial populations. Thus, Rab11 is specific in terms of increasing sleep through this population, as sleep-promoting effects of other Rabs likely map to other glia. The similarity of increased sleep may therefore suggest sleep functions of Rabs in other glial populations.

Concerning the discrepancy between RepoG4 and RepoGS sleep effects of Rab11CA and DN, it is difficult to compare as Rab11 DN is lethal with constitutively expressed Gal4s, while CA is not. As we have stated above, it is possible that CA and DN mutations do not really have opposing effects at the cellular level. In this case one might argue that CA is simply a weaker blocking mutation than DN. Nevertheless, Repo>Rab11CA flies do not appear to be sick and show intact general integrity of the BBB (Figure 4—figure supplement 2).

18) In Figure 4—figure supplement 1B- are the effects of Rab3, Rab9 and Rab11 that were significant in Figure 4A (without RU486) still significant in the Figure 4—figure supplement 1B (with RU486)? The significance stars in Figure 4—figure supplement 1B are shown only for Rab1, Rab5, Rab27 and Rab30.

Our standard for selection, as further clarified in the manuscript, is that all available DNs be significantly different from all available CAs. By this measure Rab3, 9 and 11 are different in the no-RU condition, and 1, 5, 27 and 30 are such in the RU condition, hence the difference in stars. Nevertheless, if we look at individual comparisons for the aforementioned Rabs in the RU+ condition, we find that the lone DN line of Rab3 is significantly different from one (of two) CA lines, a single DN line for Rab9 is significantly different from both CAs, and a single DN line for Rab11 is significantly different from both CAs.

19) The apparent contradiction between Figure 1, Figure 2, Figure 3, Figure 4 and Figure 5 should be more clearly explained. Figure 5 shows that SD (which increases sleep pressure) causes increased endocytosis, but, Figure 1, Figure 2, Figure 3 and Figure 4, show that inhibiting endocytosis actually increases sleep, suggesting that endocytosis in BBB promotes wakefulness. An explanation that connects the two observations should be included.

To address this point we have built upon a relevant line in the Discussion section. In short, we propose that endocytosis in the BBB is a function of sleep and serves a homeostatic need. As shown in Figure 5, endocytosis is enhanced during times of baseline sleep, or during rebound sleep induced by sleep deprivation. If endocytosis during sleep fulfills a homeostatic role, it would follow that endocytosis would decline as sleep continues (see Figure 5B), just as sleep pressure does. In the experiments where we inhibit endocytosis at the surface glia, we believe we have created a situation where the potential homeostatic function of endocytosis cannot be accomplished, hence these flies may not be able to dissipate sleep pressure, and therefore have perpetually increased sleep.

20) Are the changes in endocytosis that correlate with sleep state specific only to surface glia? Are there changes in endocytosis in other populations of glial cells, as function of sleep?

Our interest in endocytosis in the surface glia stems from our findings that Shi expression in these populations produces changes in sleep behavior. This does not exclude the possibility of sleep-dependent changes in endocytosis in other glial populations (or even neuronal populations). For example, phagocytic engulfment may be altered in astrocytes by sleep deprivation (Bellisi et al., 2017).